# Current challenges in modeling far-range air pollution induced by the 2014–15 Bárðarbunga fissure eruption (Iceland)

Marie Boichu[1], Isabelle Chiapello[1], Colette Brogniez[1], Jean-Christophe Péré[1], Francois Thieuleux[1], Benjamin Torres[1], Luc Blarel[1], Augustin Mortier[2], Thierry Podvin[1], Philippe Goloub[1], Nathalie Söhne[3], Lieven Clarisse[4], Sophie Bauduin[4], François Hendrick[5], Nicolas Theys[5], Michel Van Roozendael[5], and Didier Tanré[1]

[1]Laboratoire d'Optique Atmosphérique, Université Lille 1, CNRS/INSU, UMR8518, Villeneuve d'Ascq, France
[2]Norwegian Meteorological Institute, Oslo, Norway
[3]Atmo Nord–Pas-De-Calais, Lille, France
[4]Spectroscopie de l'Atmosphère, Service de Chimie Quantique et Photophysique, Université Libre de Bruxelles, Brussels, Belgium
[5]Belgian Institute for Space Aeronomy (BIRA-IASB), Brussels, Belgium

*Correspondence to:* Marie Boichu, marie.boichu@univ-lille1.fr

**Abstract.** The 2014–15 Holuhraun lava-flood eruption of Bárðarbunga volcano (Iceland) emitted prodigious amounts of sulfur dioxide into the atmosphere. This eruption caused a large-scale episode of air pollution throughout Western Europe in September 2014, the first event of this magnitude recorded in the modern era. We gathered chemistry-transport simulations and a wealth of complementary observations from satellite sensors (OMI, IASI), ground-based remote sensing (lidar, sunphotometry, differential optical absorption spectroscopy) and ground-level air quality monitoring networks to characterize both the spatial-temporal distributions of volcanic $SO_2$ and sulfate aerosols as well as the dynamics of the planetary boundary layer. Time variations of dynamical and microphysical properties of sulfate aerosols in the aged low-tropospheric volcanic cloud, including loading, vertical distribution, size distribution and single scattering albedo, are provided. Retrospective chemistry-transport simulations at low horizontal resolution (25 km × 25 km) capture the correct temporal dynamics of this far-range air pollution event but fail to reproduce the correct magnitude of $SO_2$ concentration at ground-level. Simulations at higher spatial resolution, relying on two nested domains with finest resolution of 7.3 km × 7.3 km, improve substantially the far-range vertical distribution of the volcanic cloud and subsequently the description of ground-level $SO_2$ concentrations. However, remaining discrepancies between model and observations are shown to result from an inaccurate representation of the planetary boundary layer (PBL) dynamics. Comparison with lidar observations points out a systematic under-estimation of the PBL height by the model, whichever the PBL parameterization scheme. Such a shortcoming impedes the capture of the overlying Bárðarbunga cloud into the PBL at the right time and in sufficient quantities. This study therefore demonstrates the key role played by the PBL dynamics in accurately modeling large-scale volcanogenic air pollution.

# 1 Introduction

On a local scale, the detrimental impact of volcanic gas, acid aerosol and ash emissions on the atmospheric environment (air pollution, rain acidification) and terrestrial ecosystems (soil, vegetation, groundwater, animals and humans) is well recognized (Delmelle, 2003; Hansell and Oppenheimer, 2004; Longo et al., 2008; van Manen, 2014; Horwell and Baxter, 2006; Ayris and Delmelle, 2012). However, volcanic sulphur-rich degassing can also generate air pollution events on a continental scale. Historical archives record evidences of long-range transport of acidic gases and aerosols from the 1783–84 Laki lava flood eruption (Iceland) up to Western and Central Europe (Thordarson and Self, 2003). Concomitantly, an abnormally high human mortality rate was observed not only in Iceland but also in Western Europe (Thordarson and Self, 2003; Witham and Oppenheimer, 2004; Grattan et al., 2005; Oppenheimer, 2011). In the specific case of the Laki eruption, it is difficult to draw a distinction between the respective impacts of volcanogenic air pollution and severe meteorological conditions, as extremes of heat and cold (which may have been partly caused by the eruption itself) occurred concurrently with the eruption (Oppenheimer, 2011). Nevertheless, there is little doubt that a Laki-style eruption would cause severe health hazards leading to an excess mortality rate at a continental scale (Schmidt et al., 2011). Obviously, at the time of the Laki eruption, only sparse observations on meteorological conditions (Yiou et al., 2014) and dispersed volcanic compounds (Thordarson and Self, 2003) were available, which hinders a thorough test of our ability to accurately model the dispersal of the prodigious emissions of volcanic $SO_2$ toward remote regions.

The long-lasting Holuhraun lava flood eruption (Aug 2014–Feb 2015) within the Bárðarbunga volcanic system (Iceland), hereafter called "Bárðarbunga eruption", allows for quantitatively assessing the far-range impact of a volcanic eruption on air quality. Even if of lesser magnitude than Laki (about one order of magnitude smaller in terms of emitted lava and sulphur degassing budgets (Gíslason et al., 2015)), the 6 month-long Bárðarbunga eruption continuously emitted abundant quantities of $SO_2$ into the lower troposphere reaching 11–12 Mt according to petrological estimates and ground-based UV-DOAS (Differential Optical Absorption Spectroscopy) observations (Gíslason et al., 2015). Whereas $SO_2$ air pollution is generally of anthropogenic origin, mainly associated with the combustion of sulfur-rich fossil fuels or with mining activities, the Bárðarbunga emissions have exceeded the budget of $SO_2$ emitted annually by all 28 state members of the European Union (4.6 Mt in 2011 (European Environment Agency, 2014)). Whereas $SO_2$ was released in large quantities, Bárðarbunga ash emissions were limited and therefore did not disturb air traffic, unlike the ash-rich 2010 Eyjafjallajökull and 2011 Grimsvótn icelandic eruptions. Nevertheless, Bárðarbunga volcano triggered an event of volcanogenic air pollution unprecedented in Europe in the modern era. Such pollution necessitated exceptional civil protection measures in Iceland(Gíslason et al., 2015). Indeed, high ground-level concentration of $SO_2$ and sulfate aerosols, mainly issued from the conversion of $SO_2$ in the atmosphere, is harmful to human health. $SO_2$ concentrations up to 9000–21000 $\mu$g.m$^{-3}$ were recorded in Iceland at a hundred kilometers from the eruption site, i.e. $\sim$60 times the hourly exposure limit value of 350 $\mu$g.m$^{-3}$ fixed by World Health Organization (WHO) (Gíslason et al., 2015).

The Bárðarbunga cloud travelled most often from the eruption site toward high latitudes, beyond the Arctic Polar Circle (Fig. 1 and supplementary material ofMcCoy and Hartmann (2015)). However, owing to favourable meteorological condi-

tions, the volcanic cloud was transported toward Western Europe in September 2014. This event fueled a far-range pollution event in SO$_2$ and particles which was recorded, without being exhaustive, in Fenno-Scandinavia (Ialongo et al., 2015; Grahn et al., 2015), Ireland, UK, the Netherlands (Schmidt et al., 2015) and France (Boichu, 2015). Contrary to stratospheric sulfate aerosols, few studies have allowed to fully determine microphysical properties of volcanic sulfates in aged tropospheric

plumes (e.g. Bukowiecki et al. (2011) in the upper-tropospheric Eyjafjallajökull cloud in 2010), due to the difficulty to isolate the signature of sulfate from co-existing meteorological clouds and/or aerosols of a different nature. Here, we use a wealth of complementary observations from in-situ ground-level sampling (SO$_2$ and particulate matter), ground-based remote sensing (lidar, sun-photometry, UV-DOAS spectroscopy) available in Belgium and France, and satellite sensors (OMI and IASI) to characterize the distribution of volcanic SO$_2$ and sulfate aerosols as well as the dynamics of the planetary boundary layer

(PBL). Both dynamical and microphysical properties of sulfate aerosols in the aged low-tropospheric Bárðarbunga cloud are provided.

We take advantage of this exceptional panel of observations to quantitatively examine and test our modeling ability to retrospectively reproduce the volcanogenic event of long-range air pollution taking place in late September 2014. While also relevant for industrial accident studies, such an exercise is critical to get prepared to accurately forecast a future large-scale

episode of volcanogenic air pollution. Indeed, geological records indicate that Laki-style high-discharge lava flood eruptions, which emit huge amounts of sulphur compounds into the atmosphere, can occur in Iceland a few times per millennium (Thordarson and Larsen, 2007).

## 2  Methodology

### 2.1  Satellite/ground-based remote sensing and in-situ sampling

According to IASI observations of the altitude of Bárðarbunga SO$_2$ near Iceland (Fig. 1), the injection height is lower than 4-5 km. Consequently, the Level-2 product of the ultraviolet-visible OMI/Aura satellite sensor for the SO$_2$ total column (NASA GES DISC, 2016) is mostly preferred to hyperspectral infrared IASI/Metop observations whose sensitivity decreases below 5 km. In addition, the center of mass of the SO$_2$ cloud is assumed to be within the PBL. North-south gaps in snapshots of the SO$_2$ cloud result from the so-called 'row anomaly' of OMI detector (www.knmi.nl/omi/research/product/rowanomaly-

background) which alters radiance data at all wavelengths for particular viewing directions. In a complementary manner, IASI captures on 21 September the front of the SO$_2$ cloud which is largely missed by OMI due to the 'row anomaly'. The major advantage of IASI is that it can track the altitude of SO$_2$ (Clarisse et al., 2014), even from moderate eruptions (Boichu et al., 2015).

[Figure 1 about here.]

A continuously-operating ground-based platform, with various remote sensing instruments, is installed on the roof of the Laboratoire d'Optique Atmosphérique in Lille-Villeneuve d'Ascq (northern France) and allows for tracking aerosols. It includes a micro-pulse CIMEL lidar measuring the radiation elastically backscattered by atmospheric particles and molecules at

532 nm. The BASIC algorithm (Mortier et al., 2013) allows for determining the vertical distribution of atmospheric particles over Lille as a function of time and distinguishing meteorological clouds from aerosols which are the focus of our study. A high load of low-tropospheric aerosols, lying at an altitude below 1.2 km, is highlighted and suspected to be partly or mostly of volcanic origin (Fig. 2).

Lidar observations are also used here to follow the PBL dynamics. The PBL is detected by applying a wavelet covariance transform to lidar backscatter profiles averaged over 20 minutes (Brooks, 2003). The PBL top is defined as the location of the maximum in the covariance profiles. As low-level meteorological clouds may disturb PBL height retrieval, a filter is applied so as to provide only cloud-free heights.

[Figure 2 about here.]

Complementary to lidar observations, the retrieval of ground-based sunphotometric observations, which are performed at two 80 km-distant sites (Lille and Dunkerque/Dunkirk), allows for identifying and isolating the signature of Bárðarbunga aerosols from other atmospheric particles transported over the north of France, such as cirrus particles here. Due to frequent cloudy conditions, time variations of vertically integrated aerosol properties derived from level-1.0 (not cloud-screened) and 2.0 (cloud-screened and quality assured) sunphotometric data from the AERONET network (Holben et al., 2001) are exploited

using different inversion algorithms and a two-site approach (Lille and Dunkerque) (Fig. 3). Fine (sub-micron) and coarse (super-micron) aerosol optical depths (AOD) at 500 nm are retrieved using spectral deconvolution algorithm (SDA) applied on AOD within the range 340 to 1640 nm (O'Neill et al., 2003) (Fig. 3). A sharp and significant increase in the fine mode AOD is highlighted in the early afternoon of 21 September (Fig. 3), which will be shown later (Section 3.2) to correspond to the arrival of the Bardarbunga cloud over France. A persistent fine mode is then observed in the following days. Volume

size distribution (VSD) of volcanic aerosols are determined using two different inversions: AERONET (version 2) standard algorithm which requires cloud-free almucantar observations (Dubovik and King, 2000; Dubovik et al., 2006) and recently developed GRASP (Generalized Retrieval of Aerosol and Surface Properties) code (Dubovik et al., 2014). Over the period of study, there is only one almucantar in Lille fulfilling AERONET level-2.0 requirements (on 23 September). Therefore, for the other two days, VSD is retrieved using GRASP: on 21 September, GRASP inverts a (manually inspected) cloud-free principal

plane as in AERONET almucantar standard inversion (Torres et al., 2014); on 22 September, direct sun (DS) measurements (available without information of sky radiances) are inverted. For this latter inversion of DS observations, we assume VSD to be a bi-modal lognormal function and optical properties (i.e. refractive index and sphericity parameter) identical to those retrieved from the almucantar on 23 September. The consistency of these algorithms and strategies is shown in Fig. 4. Using a multi-site approach (ie., including AERONET VSD determined in neighbouring site of Dunkerque), the influence on the fine mode of

cirrus co-existing with Bárðarbunga aerosols on 22 September in Lille is evidenced (Bottom of Fig. 4). $SO_2$ modeling in Fig. 8 shows that the volcanic cloud passes over Dunkerque (close to Calais) a few hours before Lille. Therefore, the similarity of fine-mode components retrieved at Lille and Dunkerque indicates that cirrus in Lille weakly influence the fine-mode, which is in turn mainly associated with volcanic sulfate aerosols in this specific case.

[Figure 3 about here.]

[Figure 4 about here.]

Daily ground-based MAX(Multi-AXis)-DOAS observations in the ultraviolet are performed by BIRA-IASB in Brussels-Uccle (Belgium) and provide time series of $SO_2$ column amounts during daylight hours (time step of $\sim$ 12 min). The instru-
ment is described in Gielen et al. (2014) while retrieval method and settings can be found in Wang et al. (2014).

Ground-level concentrations of $SO_2$ and particles are routinely measured in France by a network of ground stations managed by accredited associations responsible for air quality monitoring. For this study, AIRPARIF provided observations at Neuilly-sur-Seine (near Paris) and Atmo Nord–Pas-de-Calais at Calais and Lille-Fives (northern France). Ground-level $SO_2$
concentrations are monitored by ultraviolet fluorescence with a time step of 15 min. Mass concentration of particulate matter, with diameters less than 2.5 $\mu m$ (PM2.5) and 10 $\mu m$ (PM10), are measured by TEOM-FDMS (Tapered Element Oscillating Microbalances with Filter Dynamics Measurement Systems) (time step of 15 min) or by RST (Regulated Sampling Tube) beta gauge automated air monitors (time step of 2 hours) which account for both volatile and non-volatile PM fractions.

## 2.2   Meteorological and chemistry-transport models

The atmospheric dispersion of volcanic $SO_2$ is described using the CHIMERE Eulerian regional chemistry-transport model (CTM) (Boichu et al., 2013, 2014, 2015). The model accounts for various physico-chemical processes affecting the $SO_2$ released in the atmosphere, including transport, turbulent mixing, diffusion, dry deposition, wet scavenging and gas-/aqueous-phase chemistry. However, the conversion of $SO_2$ to sulfate aerosols is not implemented in this study to avoid uncontrolled influence of uncertainties on the numerous factors governing this process in a volcanic cloud as they have not been specifi-
cally validated for application to a volcanic plume. Not accounting for this process leads to underestimate the actual volcanic emissions. CHIMERE CTM is driven by meteorological fields from the Weather Research and Forecasting (WRF) model (Skamarock et al., 2008), which is forced by NCEP (National Centers for Environmental Prediction) reanalysis data on a 6-h basis (Kalnay et al., 1996). The spin time up of WRF simulations is of five days. WRF meteorological fields have a 25 km $\times$ 25 km horizontal grid and 30 hybrid sigma-pressure vertical layers extending up to $\sim$ 19 km above sea level (a.s.l.). The dynamics
of the PBL is described by the Yonsei University (YSU) parameterization scheme, which is the most widely used scheme implemented in WRF (Hong et al., 2006). It consists of a first-order, non-local scheme with an explicit entrainment layer and a parabolic K-profile in an unstable mixed layer. The calculated PBL height is then used as an input to CHIMERE. CHIMERE simulations are performed over the period 19–24 September 2014 on a large area extending from North of Greenland down to Spain. CHIMERE CTM has the same horizontal resolution as WRF but a finer vertical resolution with 29 hybrid sigma-pressure
vertical layers extending up to 150 hPa ($\sim$ 13 km a.s.l.).

$SO_2$ emissions are poorly known. For simplicity, we model the source term as a step-function in time with an amplitude of 4700 t.h$^{-1}$, which roughly corresponds to peak values of the $SO_2$ flux retrieved from ground-based UV-DOAS spectroscopy (Gíslason et al., 2015). $SO_2$ is released along a Gaussian profile with a full width at half maximum of 100 m. Time of release

and altitude of emissions are found by trial and error so as to reproduce by visual inspection first-order features of satellite and ground-level $SO_2$ observations. As represented in Fig. 5, we find that two step-functions at (1) 1 km a.s.l. from 19 September 2014 12:00 UT until 24 September 2014 00:00 UT and (2) at 4 km a.s.l. from 20 September 2014 12:00 UT until 24 September 2014 00:00 UT, are sufficient to fit the two-wave behaviour of the Bárðarbunga cloud (Fig. 6). This upper injection height is consistent with IASI level 2 products of $SO_2$ altitude, which captured the Bárðarbunga $SO_2$ cloud in the vicinity of the source on 19, 20, 22 and 23 September 2014 (Fig. 1). Accordingly, this source term is not intended to reflect the full complexity of the actual emissions of Bárðarbunga but rather captures only the $SO_2$ parcels traveling toward Western Europe.

[Figure 5 about here.]

## 3   Results

### 3.1   Large-scale dispersal of $SO_2$ toward Europe

According to OMI satellite observations, the model reproduces the correct timing of $SO_2$ arrival in northern Scotland on 20 September 2014 descending down to the south-western coast of England on 21 September (Fig. 6). $SO_2$ observed to the north of 60°N and to the east of 5°W on 21 September is absent from simulations because these emissions, which were released at a high altitude (above 8 km a.s.l according to IASI images of the $SO_2$ altitude in Fig. 1), are not accounted for in the model source term as they are first transported toward Arctic then dispersed toward Fenno-Scandinavia, i.e. out of our domain of interest (i.e Western Europe). The model indicates a first $SO_2$ wave (wave 1 in Fig. 6 and movie of the modeled dispersal of the Bárðarbunga $SO_2$ cloud in Supplementary Material) reaching Belgium and northern France on 21 September, which cannot be confirmed by OMI observations, hampered above France due to north-south gaps resulting from detector 'row-anomaly'. It is however captured by IASI (inset in Top of Fig. 6), which also indicates that this wave travels at a low altitude below 5 km a.s.l. (Fig. 1). This first wave is then pushed and dispersed toward the Atlantic Ocean on 22 September. Note that some traces of $SO_2$ detected by OMI over mainland Europe on 22 September are not associated with wave 1 but likely result from emissions released and then transported at high altitudes (above 8 km a.s.l.) toward Fenno-Scandinavia according to IASI images of the volcanic $SO_2$ altitude (Fig. 1). However, these emissions, as stated earlier, are not accounted for in the model. As a result, these traces of $SO_2$ over mainland Europe cannot be reproduced by simulations. On 22 September, both model and observations depict a north-south elongated $SO_2$ cloud reaching France for the second time (wave 2 in Fig. 6 and movie in Supplementary Material). Modeled $SO_2$ column amounts are in agreement with OMI $SO_2$ loading. However, the absence of $SO_2$ above Fenno-Scandinavia in the OMI image contradicts the model. This inconsistency may result from inaccuracies of prescribed altitudes of $SO_2$ injection or of meteorological forcing of the model. On 23 September, the model shows the arrival of $SO_2$ above Norway/Sweden, after a long transport from Iceland up to northern Greenland. On the same day, model and observations both indicate some dispersed remnants (although of different intensity) of the second $SO_2$ wave having hit western Europe above western France and southern UK.

[Figure 6 about here.]

## 3.2 Arrival of the volcanic cloud in the far-range lower troposphere

The precise timing of arrival of the Bárðarbunga cloud in the French lower troposphere on 21 September 2014 is deduced from the synergetic analysis of volcanic $SO_2$ modeling as well as observations from ground-based lidar and sunphotometers which remotely sense aerosols, on a continuous basis, above Lille-Villeneuve d'Ascq. Sunphotometry indicates the arrival of fine
mode aerosols between 12:00 and 15:30 UT on 21 September (Fig. 7-a1), presumably sulfate aerosols formed from volcanic $SO_2$ in the atmosphere, according to their high single scattering albedo ($\sim 0.98$) derived from AERONET inversions indicative of non- or weakly-absorbing aerosols. While fine mode AOD values remain below 0.1 at midday in Lille on 21 September, the arrival of volcanic sulfate aerosols marks an increase in AOD with values ranging between 0.3 and 0.45 in the afternoon (Fig. 3). Principal plane inversion also provides volume size distribution (Fig. Fig. 7-a2), with an effective radius ($r_{eff}$) of
these sulfate of 0.21 $\mu$m (mean volume radius $r_v$ of 0.26 $\mu$m). Simultaneously, lidar active observations, which characterize the temporal evolution of aerosol vertical distribution, indicate the presence above Lille of aerosols at 2 km a.s.l., with a decreasing altitude with time (Fig. 7-b). This behavior of aerosols coincides with the temporal decrease of the modeled altitude of the most concentrated layer of volcanic $SO_2$ accompanying the first $SO_2$ wave described in Section 3.1 (red line in Fig. 7-b). This common evolution evidences the co-existence of $SO_2$ and sulfate aerosols within the low-altitude volcanic cloud.
Soon thereafter, ground-level sampling in Lille records the first significant increase of $SO_2$ concentration up to $\sim 20$ $\mu$g.m$^{-3}$ (compared to background values usually close to zero at this site except when contaminated by nearby urban heating plant) followed by a first rise in particulate matter abundance up to $\sim 35$ $\mu$g.m$^{-3}$ (Fig. 7-c). Hence, these four pieces of evidence ($SO_2$ modeling, sunphotometry, lidar and ground-level air sampling) unambiguously confirm the arrival of the Bárðarbunga cloud in the French lower troposphere down to the ground in the early afternoon of 21 September.
After a period of quiescence, a second, more prolonged and intense episode of ground-level air pollution, in both $SO_2$ and particles, is recorded from 22 to 23 September in Lille (Fig. 7-c). During this second episode, the PM concentration exceeds the information and recommendation threshold prescribed by WHO of $\sim 50$ $\mu$g.m$^{-3}$, defined as the hourly running 24 hour average value. Concomitantly, sunphotometry indicates a persistent fine-mode (Fig. 7-a2) of weakly absorbing aerosols, which produce fine mode AOD values abnormally high for Lille and Dunkerque (up to $\sim 0.8$, Fig. 3). The size of Bárðarbunga
sulfate aerosols ($r_{eff}$ within 0.26–0.28 $\mu$m, $r_v$ within 0.21–0.24 $\mu$m) largely exceeds the radius characterizing typical urban aerosols in Lille ($r_{eff} < 0.2$ $\mu$m (Mortier, 2013)). This size is also larger than values reported by sparse observations of volcanic tropospheric sulfate radius at distance from the volcanic source ($r_v$ within 0.12–0.16 $\mu$m in the Eyjafjallajökull cloud (Bukowiecki et al., 2011)).

[Figure 7 about here.]

## 3.3 Far-range air pollution at ground level

Substantial increases in ground-level $SO_2$ concentration are recorded by air quality monitoring networks not only in the north end of the country but also on a broad regional scale in France. Unseen for more than a decade, this makes this event of $SO_2$ pollution exceptional (Fig. 8). Interestingly, this pollution episode strictly follows a similar temporal pattern, except for a

time lag, whichever the city of observation. As shown from the combined analysis of space-based $SO_2$ observations and CTM simulations at a large scale (Section 3.1 and movie of the modeled dispersal of the Bárðarbunga $SO_2$ cloud in Supplementary Material), this temporal behaviour results from the arrival of two successive waves of $SO_2$ reaching France from 21 to 23 September. At ground-level, air quality measurements track the progressive transport of these two waves from the north to

the center of France (blue lines in Fig. 8). $SO_2$ concentrations up to 70 $\mu$g.m$^{-3}$ are associated with the second wave, which is recorded firstly in Calais, then successively 3 hours later in Lille-Fives and 8 hours later much further south near Paris in Neuilly-sur-Seine. While the modelled time series of $SO_2$ column amounts reproduce this two-wave pattern (solid red line in Fig. 8), simulations fail in correctly describing ground-level $SO_2$ concentration as the second wave of pollution starting on 22 September is missed (dashed red line in Fig. 8).

10                                                            [Figure 8 about here.]

## 4    How to improve long-distance air quality modeling?

### 4.1    Limitations of simulations with a standard configuration

As illustrated by the broad agreement with OMI satellite data (Fig. 6 and Section 3.1), chemistry-transport simulations, with a standard configuration here, are efficient at reproducing on a continental scale the dispersion of the Bárðarbunga $SO_2$ cloud

from Iceland toward France. The temporal dynamics of far-range events of air pollution, characterized for instance in France by the arrival of two distinct $SO_2$ waves traveling from north toward the capital city in $\sim$ 8 hours, is hence well described (solid red line in Fig. 8). A good agreement is also reached between model and observations of $SO_2$ vertical column amounts (CA) by ground-based UV max-DOAS spectroscopy performed at long distance from the eruptive site in Uccle/Belgium, located less than 100 km from Lille/France (Fig. 9). Model and observations find $SO_2$ CA of the same magnitude on 19, 20, 22 and

23 September. Nevertheless, the model cannot capture the significant and abrupt $SO_2$ CA increase (up to 14 DU) of very short duration (from 14:18 to 15:07 UT) observed on 21 September by DOAS, likely due to the insufficient (hourly) time resolution of the model.

[Figure 9 about here.]

However, the model has difficulty reproducing the correct intensity of the air pollution episode in remote areas. Similar

issues have also arisen with independent modeling simulations using a Lagrangian approach forced with distinct meteorological reanalysis (Schmidt et al., 2015). Our model here completely misses the second wave of $SO_2$ at ground-level in the north of France (dashed red line in Fig. 8). This shortcoming results from an incorrect description of the vertical distribution of $SO_2$ at long distance from the eruptive site. According to lidar observations capable to detect sulfate aerosols coexisting with $SO_2$ (Section 3.2), the model mimics correctly the drop in altitude above Lille of the first $SO_2$ wave on 21 September (red line in

Fig. 7-b). This modeled wave hits the surface at about the same time as the first detection of air pollution at ground-level. But the second modeled wave, despite a similar pattern with a significant decrease in altitude with time from 6 km a.s.l., does not reach the ground and remains at an altitude $\geq$ 1.8 km above Lille on 22 September (red line in Fig. 7-b).

Issues encountered for adequately modeling far-range air pollution episodes can arise from the difficulty of simulating both the long-range transport/dispersal of volcanic compounds and the meteorological dynamics at a local scale, as the latter controls the capture and mixing of the overlying volcanic cloud in the far-range planetary boundary layer (PBL). Simulations at higher spatial resolution of both CHIMERE CTM and WRF models may help to make progress along this path. For these reasons, we explore in the next sections the impact on far-range ground-level $SO_2$ concentrations of both meteorological/chemistry-transport simulations at higher spatial resolution and of various PBL parameterization schemes in the meteorological model.

## 4.2 Improvements reached with simulations at higher spatial resolution

Meteorological and chemistry-transport simulations at higher spatial resolution require both high computation time and capacity, which challenges our current modeling capacities. We performed here WRF and CHIMERE simulations on two nested horizontal grids (Fig. 10). The larger domain extends from north of Greenland down to Spain (as the low resolution domain of simulations in Sections 2.2 and 3) with a horizontal resolution of 22 km $\times$ 22 km representing 209 $\times$ 229 grid cells. Note that this coarse resolution is nevertheless slightly higher than the low spatial resolution simulation performed on a 25 km $\times$ 25 km horizontal grid. The nested domain extends from Norway down to Central France, with a fine 7.3 km $\times$ 7.3 km horizontal resolution representing 217 $\times$ 232 grid cells in a Lambert projection. Except for one test run configured with 60 hybrid sigma-pressure vertical layers extending up to $\sim$ 19 km a.s.l. and $\sim$ 13 km for WRF and CHIMERE models respectively, most simulations are run assuming 30 vertical layers for both models, as in the simulations with a standard configuration in Section 3. The dynamics of the PBL is still described by the Yonsei University (YSU) scheme, as in Section 3.

[Figure 10 about here.]

Simulations at higher horizontal spatial resolution better resolve the long-distance transport of the Bárðarbunga $SO_2$ cloud as well as its descent over France. Especially, an earlier and faster descent of the $SO_2$ cloud over Lille is modeled on 22 September 2014, with the core of the plume reaching a significantly lower altitude than in low resolution simulations (Top panels of Fig. 11). Subsequently, a clear improvement of modeled far-range ground-level $SO_2$ concentrations is reached. Indeed, this earlier modeled descent of the $SO_2$ cloud leads to the emergence of a second peak in ground-level concentrations on 22 September (Bottom panel of Fig.11-right), which was entirely missed by simulations at low spatial resolution as mentioned in Section 4.1 (Bottom panel of Fig. 11-left).

[Figure 11 about here.]

The emergence of a second peak is also modeled at other air quality monitoring stations, i.e. Calais and Neuilly-Sur-Seine (Fig. 12). A better agreement between model and observations is also noticed regarding the timing of the first peak concentration whichever the station (Fig. 12). Note that only slight differences in ground-level concentrations were observed with simulations performed with a twice higher vertical resolution (i.e. 60 vertical layers in WRF and CHIMERE models) and are consequently not shown.

Nevertheless, although clear improvements in ground-level $SO_2$ concentrations are achieved with simulations at higher spatial resolution, both timing and intensity of the second peak concentration are not perfectly reproduced by the model (Fig. 12). The emergence of the second peak concentration is modeled late compared to measurements at Calais and Lille, and slightly too early at Neuilly-sur-Seine. The modeled intensity of this peak concentration is always under-estimated by a factor of 3 to 10 depending on the monitoring station.

These remaining discrepancies between model and observations may arise from various reasons. We explore in the next sections the impact on far-range ground-level concentrations of (i) a poor knowledge of the source emissions (assumed here to follow a simple pattern as described in Section 2.2), and (ii) an incorrect modeling of the planetary boundary layer dynamics which may prevent from correctly capturing the descending volcanic cloud down to the ground.

### 4.3 Minor role of source term variations

We do not aim to provide a detailed estimate of the source term but to show that a simple source term allows for representing the main features of this event of far-range air pollution triggered by Bárðarbunga eruption (Sections 3.1 and 3.3). Instead, we investigate here whether variations in this simple source term could explain the current discrepancies between observed and modeled far-range $SO_2$ concentrations at ground-level.

Whichever the monitoring station, we have deduced from $SO_2$ modelling that the second peak in ground-level $SO_2$ concentration results from the arrival in France of the second pulse of emissions, which are injected at the source at 4 km a.s.l. from 20 September 2014 12:00 UT until 24 September 2014 00:00 UT in low spatial resolution simulations (not shown). We vary the altitude of injection (between 3 and 7 km a.s.l.) as well as the start time of this second pulse of emissions (with a 5-hour earlier release), as both may impact the timing of the second peak concentration which is not correctly reproduced by the model (Section 4.2 and Fig.12). However, only slight modifications on far-range ground-level concentrations are obtained, unable to explain the discrepancy observed between model and observations.

Regarding the under-estimation by the model of ground-level $SO_2$ concentrations by a factor of 3 to 10 (Section 4.2), one could argue that it results from an under-estimation of the assumed $SO_2$ emission flux by a similar factor given linear processes of large-scale transport/dispersion. Volcanic emissions may present rapid temporal fluctuations of large amplitude (e.g Boichu et al. (2013)). However, despite a poor time-resolved knowledge of the Bárðarbunga $SO_2$ source relying on sparse ground-based measurements, assuming a $SO_2$ flux five time stronger would better fit far-range ground-level concentrations but would also lead to far-range $SO_2$ column amounts increased by the same amount (not shown). The latter would be in complete disagreement with $SO_2$ column amounts retrieved from satellite observations (Fig. 6) or ground-based MaxDOAS measurements performed in Belgium (Fig. 9).

As a consequence, input model parameters characterizing the Bárðarbunga $SO_2$ source (flux and altitude of injection) are shown to play a minor role on far-range ground-level concentrations over our relatively short period of study (19-24 September 2014). They do not allow us to solve the disagreement observed between model and observations. For this reason, we explore in the next section the impact of the PBL dynamics on air quality modeling.

## 4.4    Key role of the planetary boundary layer

In addition to the reference Yonsei University (YSU) scheme used in the low spatial resolution simulation, the impact on far-range ground-level concentrations of two additional PBL parameterization schemes, recently added to the WRF model, are tested: the Asymmetric Convective Model (ACM2) scheme (Pleim, 2007) as well as the improved Mellor-Yamada-Nakanishi-Niino level 3 model (MYNN 3) scheme (Nakanishi and Niino, 2006). The ACM2 scheme is a first-order, non-local closure scheme which features non-local upward mixing and local downward mixing. The MYNN 3 scheme is a second order, local closure scheme tuned to a database of large-eddy simulations.

At first glance, time series of the PBL height above Lille do not seem to vary widely with the different PBL parameterization schemes (Top of Fig. 13). We note nevertheless a less marked diurnal cycle with the MYNN3 scheme. However, these slight differences are shown to be sufficient to produce up to a ten-fold variation of the ground-level $SO_2$ concentrations (Bottom of Fig. 13).

[Figure 13 about here.]

As illustrated by Fig. 14, both timing and altitude of the encounter of the boundary layer top and the overlying volcanic $SO_2$ cloud, which may vary with the PBL parameterization scheme, strongly impact the subsequent increase in $SO_2$ concentration at ground-level some time later.

For our specific case-study, the top of the PBL encounters the overlying Bárðarbunga $SO_2$ cloud above Lille at approximately the same time on 21 and 22 September, whichever the PBL scheme (Fig. 14). After this capture of the volcanic cloud into the PBL, $SO_2$ is mixed and diffused down to the ground triggering a noticeable increase of the ground-level $SO_2$ concentration. The time delay between the capture of the volcanic $SO_2$ at the top of the PBL and its record at the ground-level is estimated of just a few hours (Fig. 14).

However, only the ACM2 scheme allows the top of the PBL to encounter the core (i.e. the most concentrated part) of the Bárðarbunga $SO_2$ cloud on 22 September 2014, with a PBL height at the time of encounter higher by just a few hundred of meters compared to other schemes (Top right of Fig. 14)). This substantial capture of volcanic $SO_2$ by the boundary layer explains why the intensity of the second $SO_2$ peak concentration is the highest with this scheme and the closest to observations (Bottom right of Fig. 14). Note that the best agreement between observations and model for the first $SO_2$ peak concentration is also reached with the ACM2 scheme. These results demonstrate the crucial importance to correctly model both the PBL

height and the vertical distribution of the overlying volcanic $SO_2$ cloud with time. Note that this latter depends on a rigorous modeling of both long-distance transport/dispersion processes and of local PBL dynamics. Indeed, the PBL scheme influences the concentration of the overlying volcanic $SO_2$ (Top of Fig. Fig. 14). We may even suspect a kind of "sucking" of the core of the volcanic cloud which seems to follow the PBL top, especially remarkable on 22 September.

[Figure 14 about here.]

Nevertheless, even if the ACM2 scheme provides the best fit to observations, none of the PBL schemes allows for precisely modeling the second $SO_2$ peak concentration with correct timing and intensity (Fig.13). This difficulty likely results from the inaccuracy of the modeled PBL height which presents marked differences with observations, whichever the PBL scheme. The
altitude of the PBL above Lille is retrieved from lidar observations and compared with model output in the bottom of Fig.15. Simulations generally underestimate the PBL height over Lille (up to 1.5 km), especially in the mornings and evenings. Such underestimation is a relatively common feature of WRF PBL schemes (Banks et al., 2015), used here to force the CHIMERE chemistry-transport model.

In a context of urban air pollution where pollutants are injected at ground-level into the atmosphere, an underestimation of
the PBL height favors an over-evaluation of the intensity of ground-level pollution. In our volcanic case-study, this may explain the overestimation by certain schemes of the $SO_2$ ground concentration increase resulting from the first $SO_2$ wave reaching the ground of Lille on 21 September in the evening (Fig. 13). However, the PBL height underestimation by the model can also prevent from correctly capturing in the PBL the second $SO_2$ wave which travels at a higher altitude than the first wave (Fig. 14). In this context, the intensity of air pollution at ground-level is under-evaluated (Bottom of Fig. 13).

In our specific case-study, our concern is that, whichever the scheme, the modeled PBL height increases too lately and too weakly compared to lidar observations in Lille, which is especially problematic in the morning of 22 September (Bottom of Fig. 15). Indeed, this discrepancy explains both the delayed modeled timing of the second peak concentration and a substantial under-estimation of its intensity (by a factor of 2-3), as the modeled boundary layer captures too late a smaller fraction of the overlying volcanic $SO_2$ than it should in reality. In other words, an earlier and higher modeled PBL height in the morning
of 22 September, as expected according to lidar observations, would lead to an earlier and stronger capture of the overlying Bárðarbunga $SO_2$ cloud at the top of the boundary layer. This would produce an earlier and stronger peak concentration at ground-level in better agreement with air quality monitoring observations.

Therefore, this case-study demonstrates the key role played by the PBL dynamics to rigorously estimate the magnitude of far-range volcanogenic air pollution.

[Figure 15 about here.]

## 5  Conclusions

The Bárðarbunga eruption provides the exceptional opportunity to carry out a modelling exercise of a far-range volcanogenic air pollution event using a broad panel of complementary measurements acquired by space and ground-based (remote sensing and in-situ) sensors.

Chemistry-transport modeling reproduces the large-scale dispersal of $SO_2$ from Iceland toward western Europe as observed from satellite OMI and IASI sensors. The synergetic analysis of $SO_2$ modeling and aerosol dynamics deduced from sunphotometric and lidar observations allows us to determine the exact timing of arrival of the volcanic cloud in the distant lower troposphere of northern France before its descent to the ground. The joint analysis of lidar measurements with the retrieval of
multi-site sunphotometric observations using recently-developed inversion algorithms also provides a full characterization of volcanic sulfate aerosol properties with time (loading, vertical repartition, size distribution and single scattering albedo).

Based on this combined analysis of volcanic $SO_2$ and sulfate aerosols, we highlight the success and the challenges in simulating far-range episodes of air pollution. We show that the air pollution triggered by the Bárðarbunga eruption in late September
2014 is characterized by the arrival to France of two distinct $SO_2$ waves. The descent of these waves down to the ground produces two substantial peak concentration recorded at different monitoring ground stations in France with a time lag of 3 to 8 hours. The specific temporal pattern of this pollution event is well described even with low (25 km × 25 km) horizontal spatial resolution simulations. However, the model faces difficulties in reproducing the correct magnitude of one of the two ground-level $SO_2$ peak concentrations.

We show that large improvements on the far-range vertical distribution of the dispersed volcanic cloud and subsequently on surface concentrations are gained with simulations carried out at higher spatial resolution. Such simulations rely on two nested horizontal grids, which include a large domain with a coarse resolution of 22 km × 22 km and a narrower domain with a fine resolution of 7.3 km × 7.3 km. High computational capacities are required given the very large extent of the area flown over
by the Bárðarbunga volcanic cloud in late September 2014, from northern Greenland down to south of France. Nevertheless, some discrepancies remain as high spatial resolution simulations do not reproduce correctly the timing of the second $SO_2$ peak concentration at ground-level (with a difference of a few hours) and the intensity of this peak is substantially under-estimated compared to observations.

The reasons for these remaining discrepancies between model and observations of far-range ground-level concentrations are investigated. Variations in the source term parameters (i.e. flux and altitude of injection) are shown to have a minor impact during the period of time of our study. However, the PBL dynamics plays a key role. Testing three parameterization schemes for the planetary boundary layer in the WRF model (YSU, ACM2 and MYNN3), a resulting ten-fold variability of surface concentrations is obtained. The ACM2 scheme provides the best fit to observations. Nevertheless, it does not perfectly repro-

duce the timing and intensity (under-estimated by a factor 3) of the second peak concentration. Lidar observations performed in Lille allows us to test the validity of the modeled PBL height time series at this location. During the morning of specific interest, the modeled PBL height increases too late and too weakly compared to observations. This shortcoming results in a too late and too weak capture of the overlying Bárðarbunga $SO_2$ cloud by the boundary layer and, subsequently, a delayed peak concentration at ground-level with an under-estimated intensity.

This case-study points out how fundamental it is to simulate accurately the PBL dynamics for modeling large-scale volcanogenic air pollution. Such difficulties will need to be overcome in order to get prepared to accurately forecast far-range air pollution episodes triggered by future eruptions releasing large amounts of toxic gases to the atmosphere.

*Acknowledgements.* M. Boichu gratefully acknowledges support from the Nord–Pas-De-Calais Regional Council for her junior research fellowship. LOA members thank the French National Research Agency for funding the VOLCPLUME project (ANR-15-CE04-0003-01), the Chantier Arctique for funding the PARCS project and the CaPPA (Chemical and Physical Properties of the Atmosphere) excellence laboratory. NASA Goddard Earth Sciences Data and Information Services Center (GES DISC) are acknowledged for providing OMI $SO_2$ data. BIRA-IASB MAX-DOAS activities at Uccle were financially supported by the projects AGACC-II (BELSPO, Brussels) and NORS (EU FP7; contract 284421). S.B. and L.C are respectively Research Fellow and Research Associate with the Belgian F.R.S-FNRS. Researchers and agencies in charge of AERONET (sunphotometry) and air quality monitoring networks (Atmo NPDC and AIRPARIF) provided invaluable observations and are gratefully thanked. Authors warmly thank Y. Derimian (LOA) for discussions on the impact of cirrus on sunphotometric retrievals. Two anonymous reviewers are thanked for their detailed and constructive reviews.

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

**List of Figures**

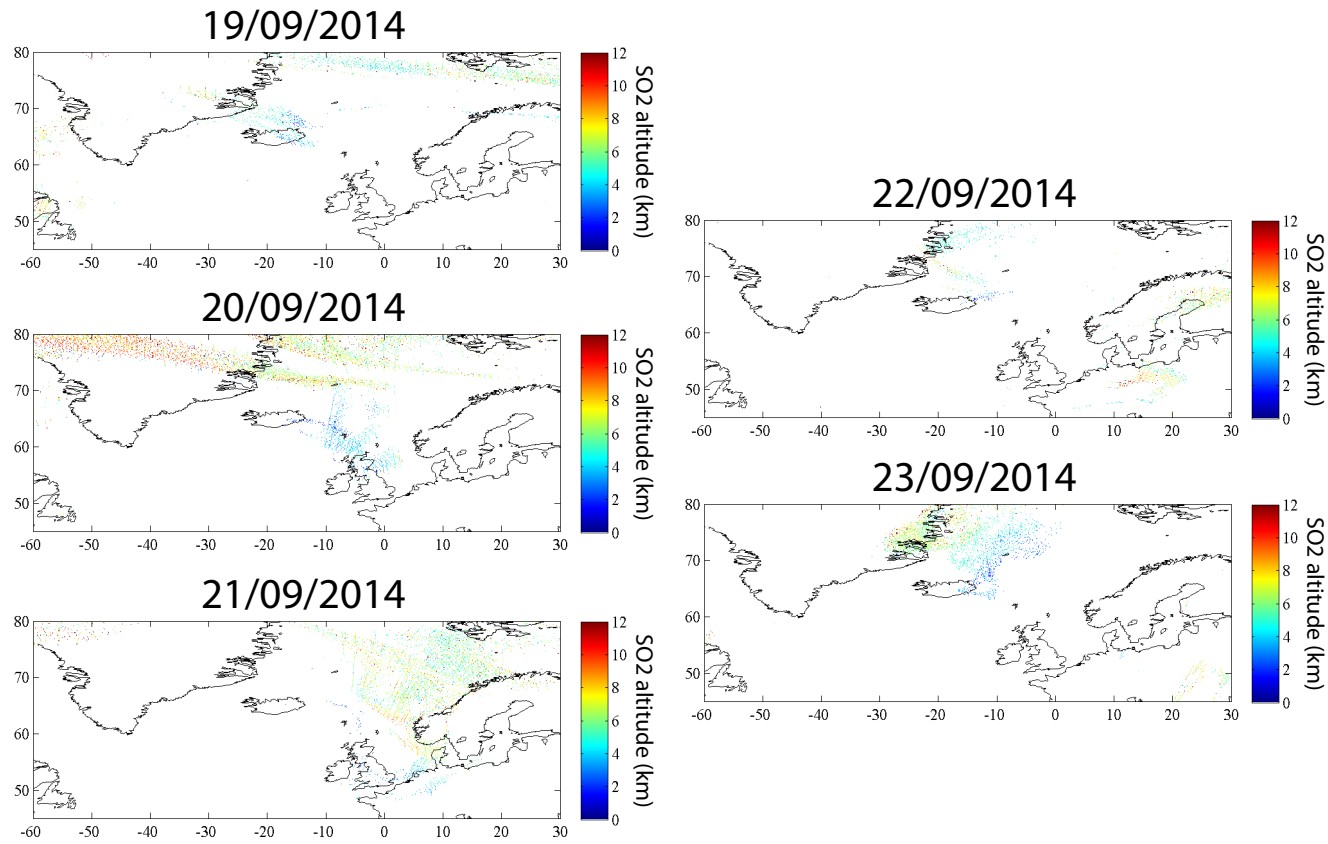

**Figure 1.** Altitude (in km a.s.l.) of Bárðarbunga SO₂ retrieved from IASI observations.

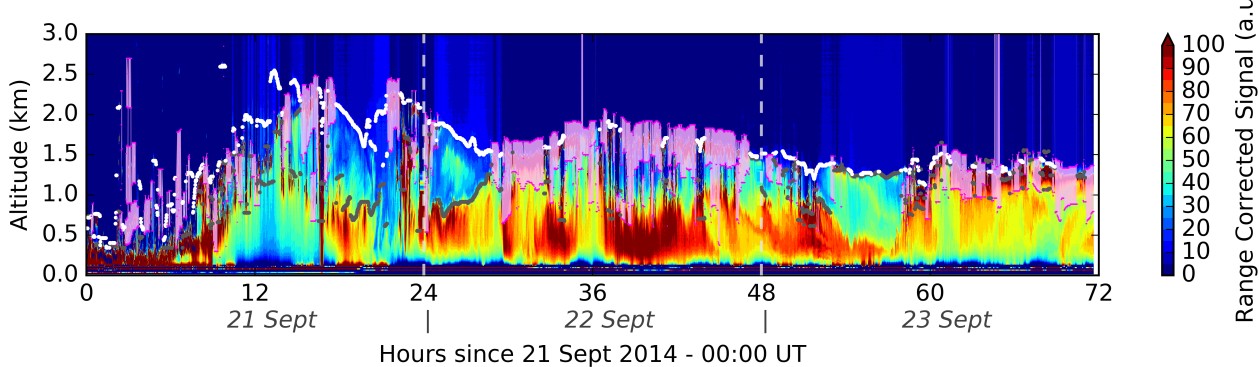

**Figure 2.** Vertical distribution of atmospheric particles detected over Lille by lidar observations performed from 21 to 23 September 2014. The BASIC algorithm detects meteorological clouds above 300 m a.s.l. (pink), the top of the aerosol layer (white) and a heavy load of low-tropospheric aerosols including volcanic particles (lying at an altitude below ∼1.2 km), which is characterized by a strong backscatter signal (yellow/red).

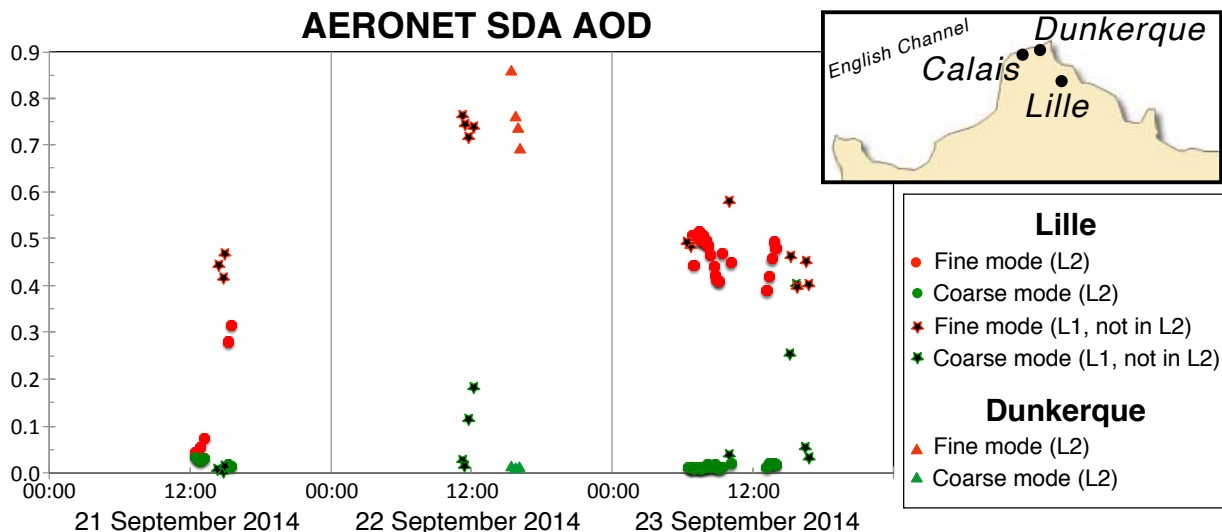

**Figure 3.** AERONET sunphotometric fine and coarse mode aerosol optical depth (AOD) at 500 nm retrieved using SDA algorithm in Lille and Dunkerque from 21 September to 23 September 2014. Map of northern France in inset.

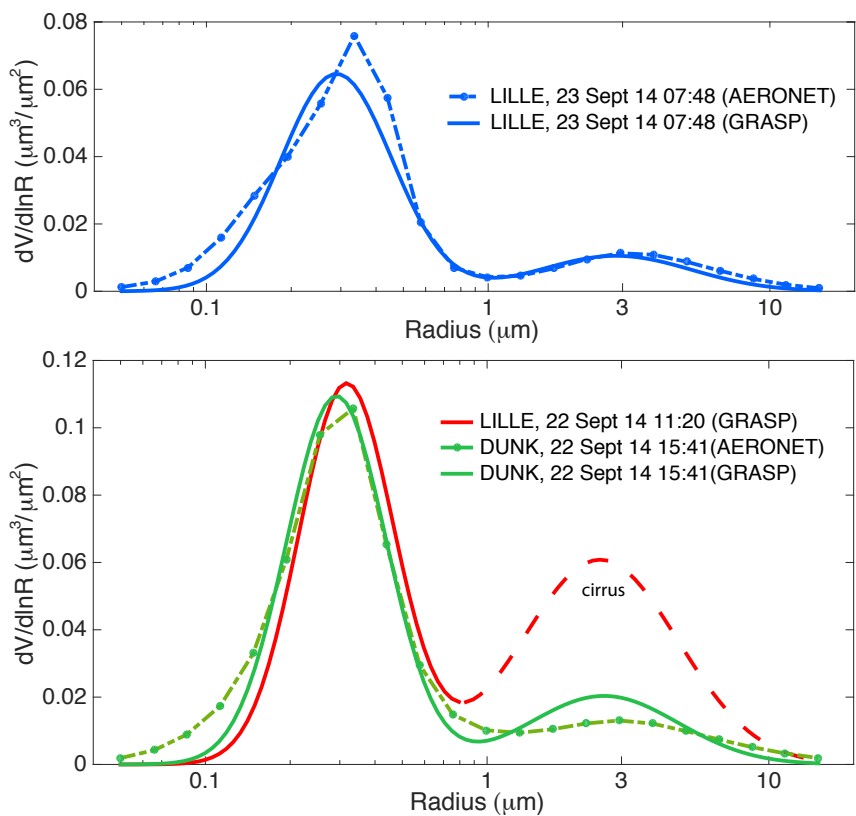

**Figure 4.** (Top) Consistency of the volume size distribution (VSD) in Lille on 23 September retrieved (dashed line) by inversion of Almucantar observations using standard AERONET inversion and (solid line) by inversion of direct sun measurements using GRASP algorithm. (Bottom) For 22 September, consistency of VSD retrieved by inversion of almucantar using standard AERONET inversion in Dunkerque (green dashed line) and by inversion of direct sun measurements using GRASP algorithm in Dunkerque (green plain line) and Lille (red line).

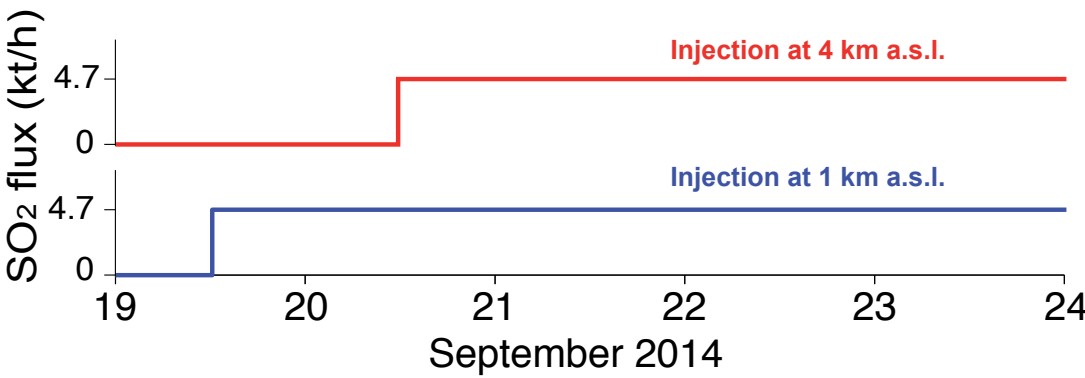

**Figure 5.** Simple source term (i.e. flux and altitude of injection of emissions as a function of time) initializing chemistry-transport simulations of the Bárðarbunga $SO_2$ cloud dispersal toward Europe in late September 2014.

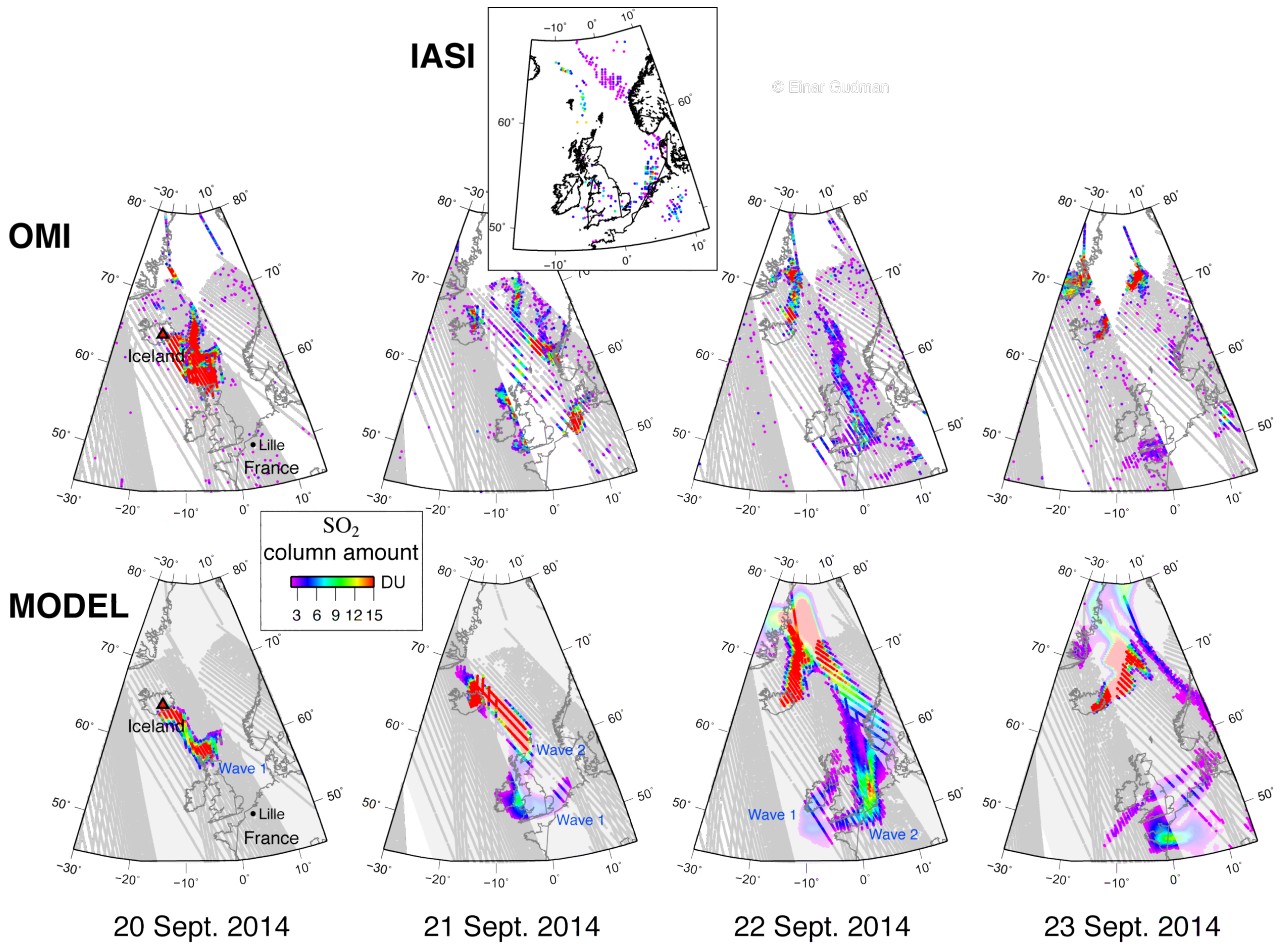

**Figure 6.** Dispersion of SO$_2$ from Bárðarbunga eruption toward Europe in late September 2014 (top) observed from satellite imagery (timeseries of OMI PBL products and IASI data in inset for 21 September) and (bottom) modeled using CHIMERE chemistry-transport model. Grey points indicate OMI column amounts < 2 DU. White zones show areas where data are not available. To facilitate the comparison between model and observations, the model is displayed transparently over zones where OMI data are not available.

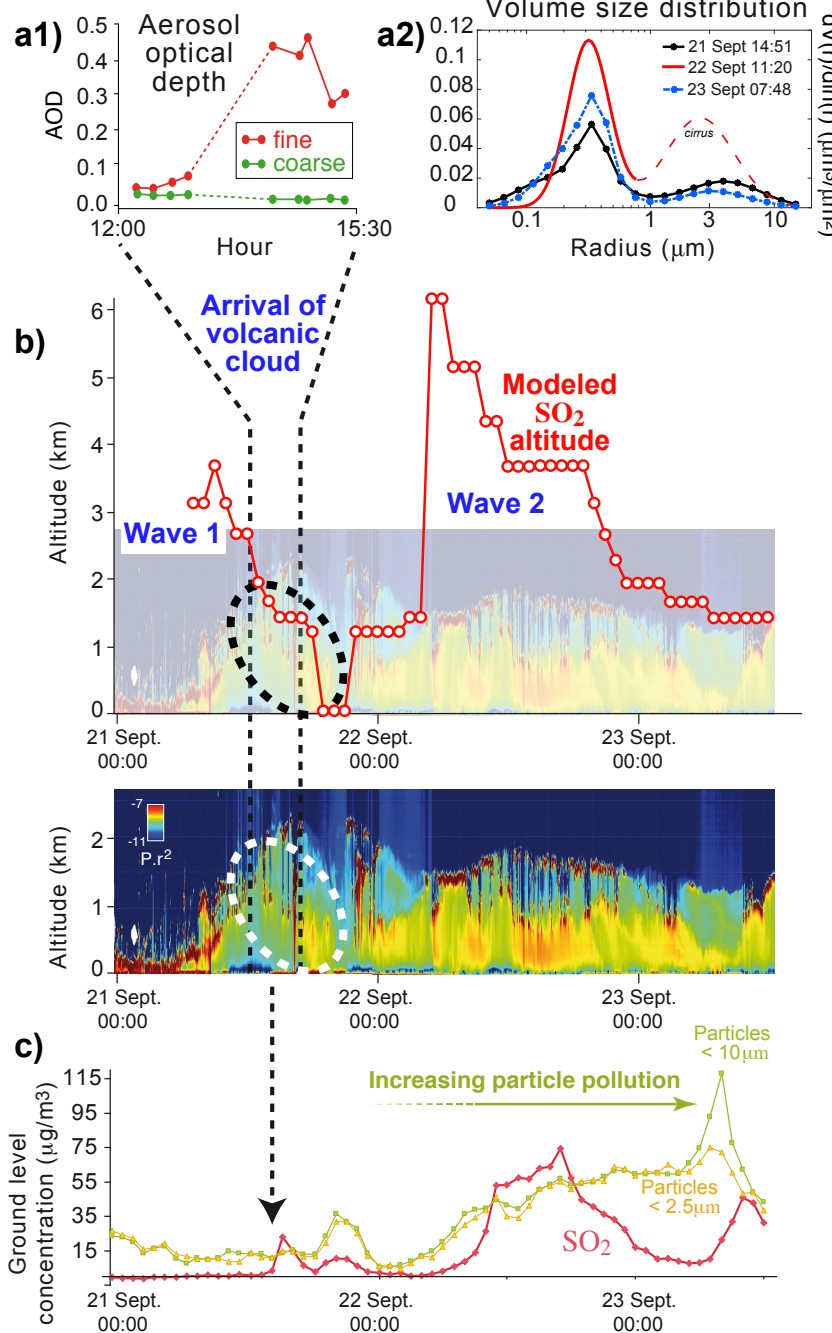

**Figure 7.** Multi-parametric observations and modeling of SO$_2$ and aerosols in Lille: (a) Retrieval of sunphotometric observations yields (a1) aerosol optical depth (at 500 nm) for coarse (green) and fine (red) mode on 21 September and (a2) time variations of aerosol volume size distribution. (b) Time series of the modeled altitude (red line) of the most concentrated layer of volcanic SO$_2$ overlaid on the lidar range-corrected backscatter signal (ln(P.r$^2$)) at 532 nm. Distinction between aerosol and meteorological clouds in lidar data is shown in Fig. 2. (c) Ground-level concentration of SO$_2$ (red) and particles (PM2.5 in yellow and PM10 in green).

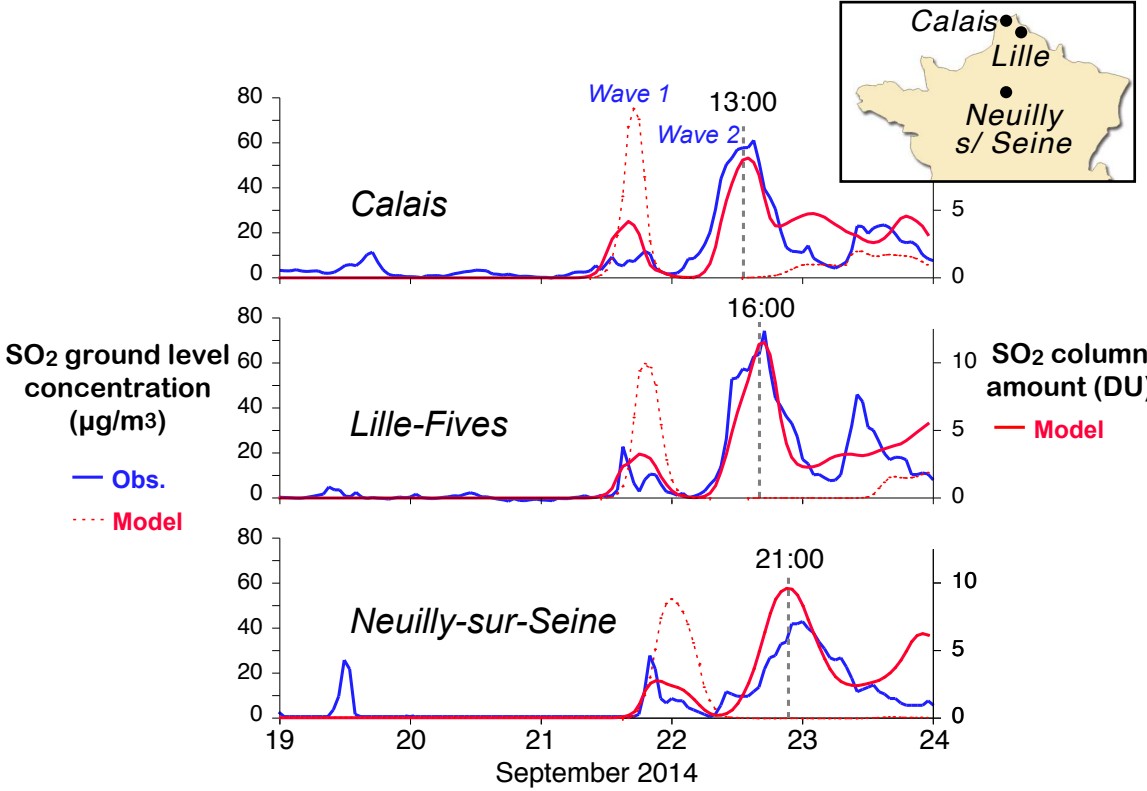

**Figure 8.** SO₂ ground-level concentration observed by air quality networks in France (blue) and modelled (dotted red), compared with modelled SO₂ vertical column amount (solid red).

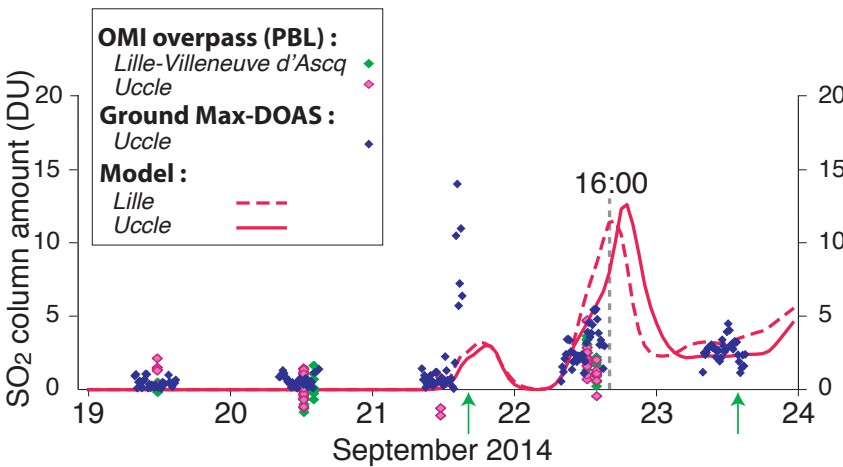

**Figure 9.** Time series of SO$_2$ vertical column amount above Uccle (Belgium) and Lille (France) from CHIMERE CT model (dashed and solid red lines for Lille and Uccle resp.), OMI PBL overpass (green and pink diamonds for Lille and Uccle resp.) and ground-based UV MAX-DOAS observations in Uccle (blue diamonds). Green arrows indicate when OMI overpasses above Lille are missing due to gaps in data related to sensor 'row anomaly'.

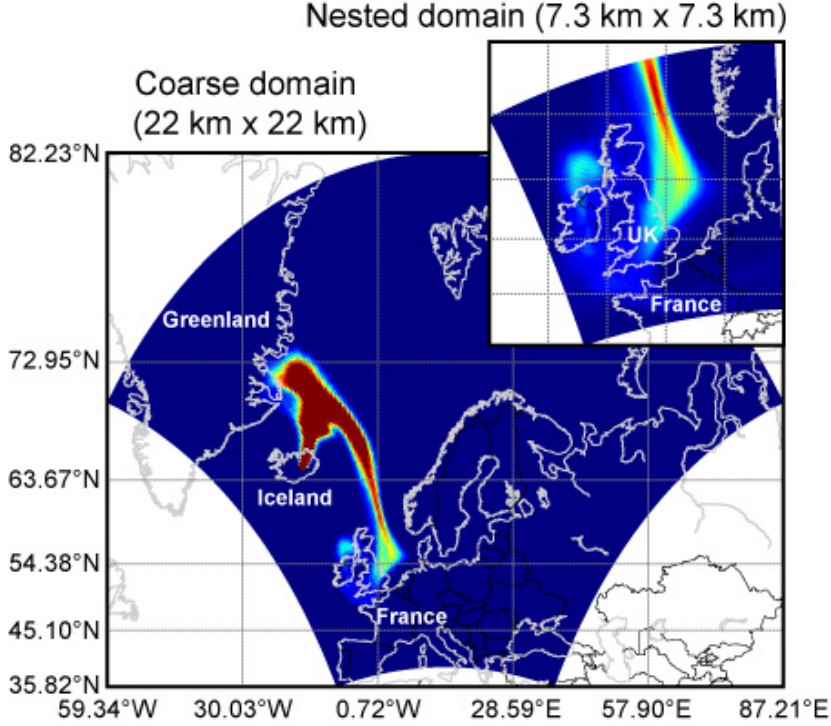

**Figure 10.** Nested horizontal domains of high spatial resolution simulations: coarse resolution (22 km × 22 km) and nested high resolution (7.3 km × 7.3 km) (in inset) domains.

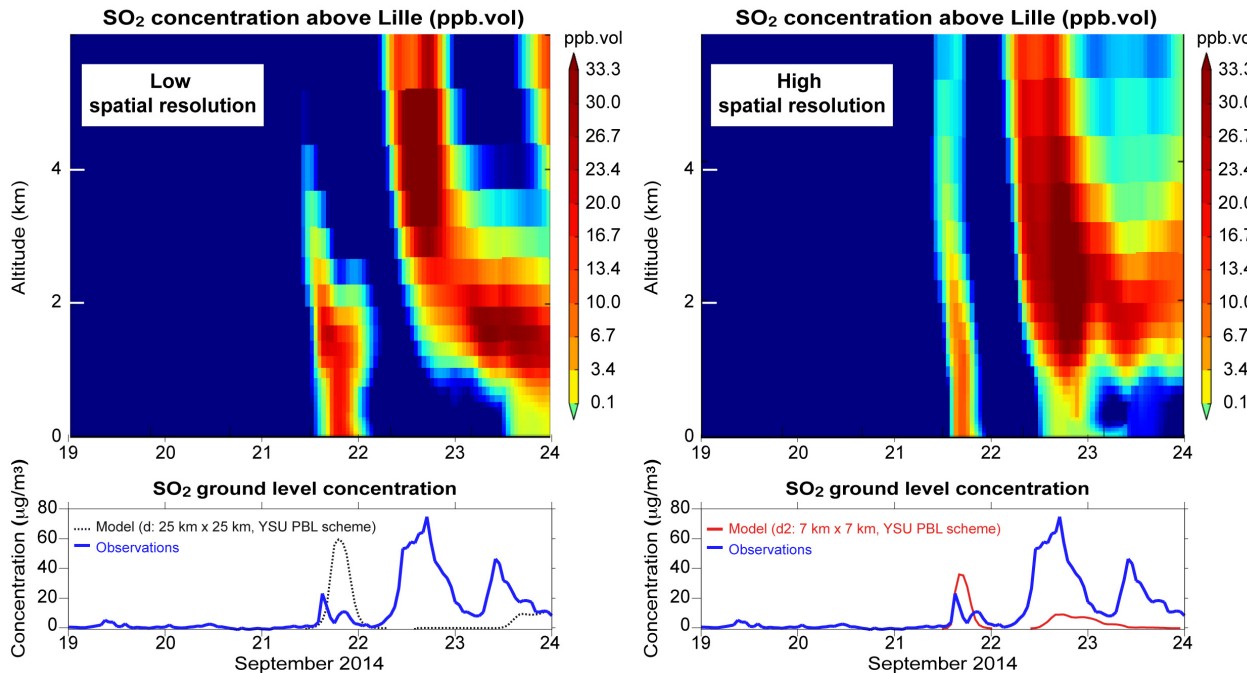

**Figure 11.** Comparison of far-range vertical distributions of SO$_2$ over Lille modeled with simulations at (left) low versus (right) high horizontal resolution (both configured with the YSU PBL parameterization scheme). Simulations at low resolution (25 km × 25 km) are performed on a single domain while simulations at higher resolution are performed on two nested domains: the largest with a coarse resolution of 22 km × 22 km and the narrowest with a fine resolution of 7.3 km × 7.3 km (Fig. 10). (Top) Modeled concentration of volcanic SO$_2$ over Lille as a function of time (X-axis) and altitude (Y-axis). (Bottom) Time evolution of the observed (blue) and modeled (red or black) SO$_2$ concentrations at ground-level.

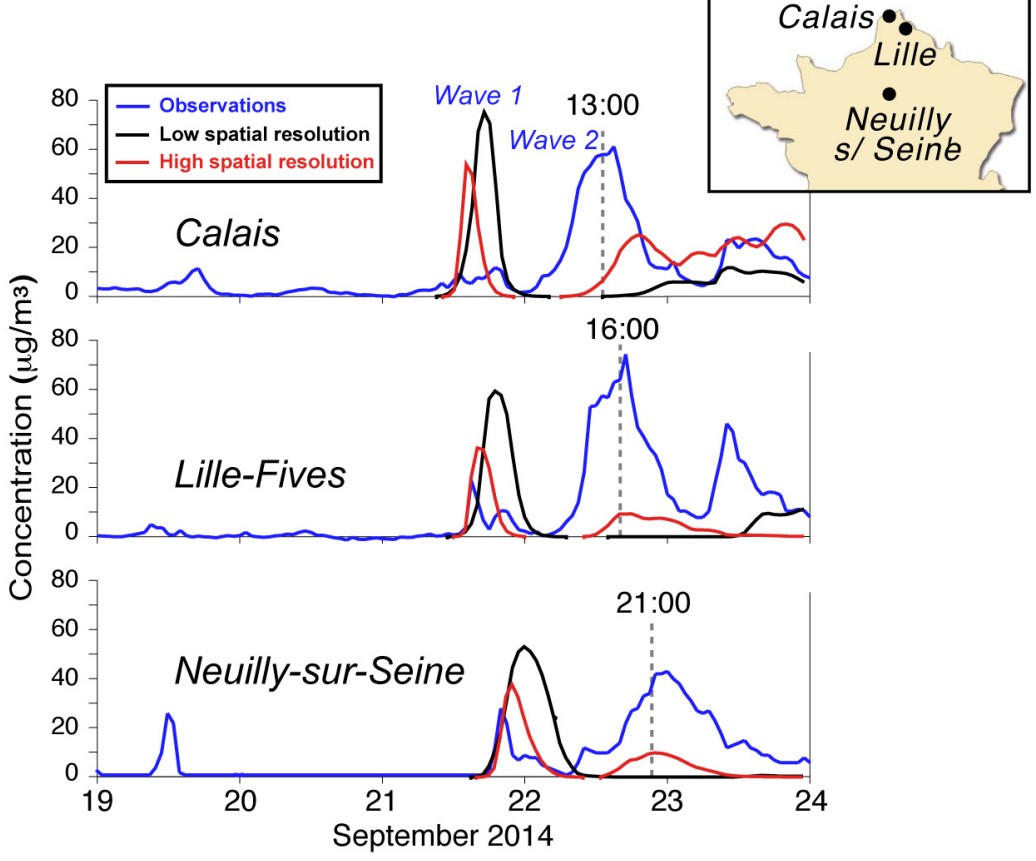

**Figure 12.** SO$_2$ ground-level concentration observed by air quality networks in France (blue) and modelled with high (red) or low (black) horizontal resolution simulations, both configured with the YSU PBL parameterization scheme.

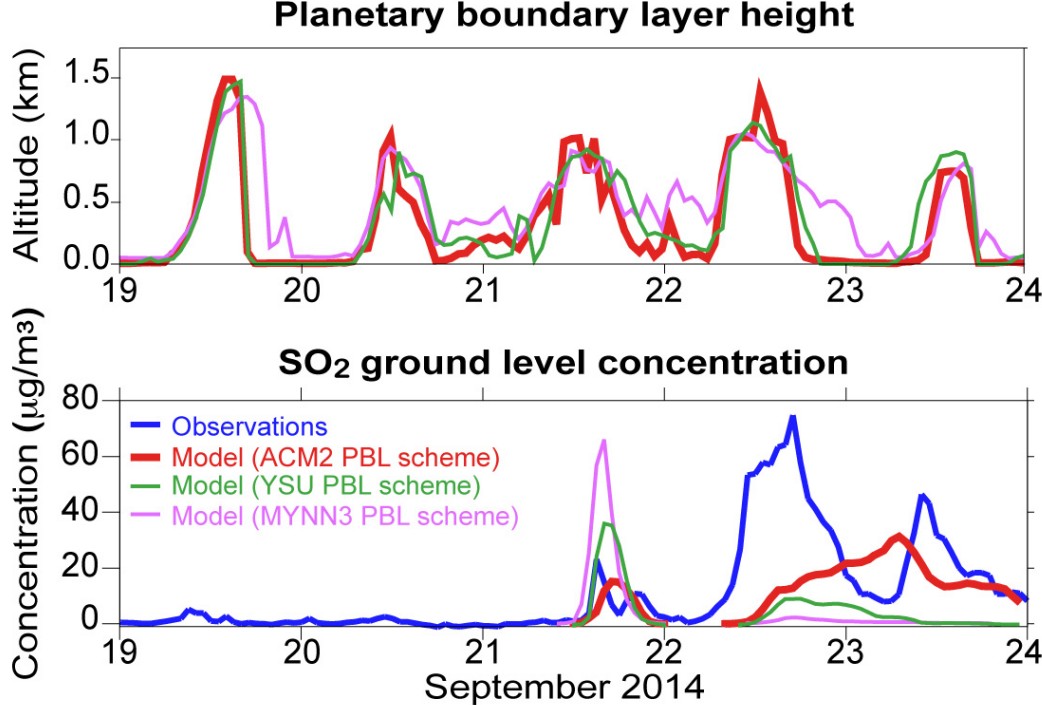

**Figure 13.** Time variations in Lille of modeled (Top) planetary boundary layer height and (Bottom) SO₂ concentration at ground-level with high spatial resolution simulations configured with YSU (green), ACM2 (red) and MYNN3 (pink) PBL parameterization schemes. The observed ground-level concentration is represented in blue.

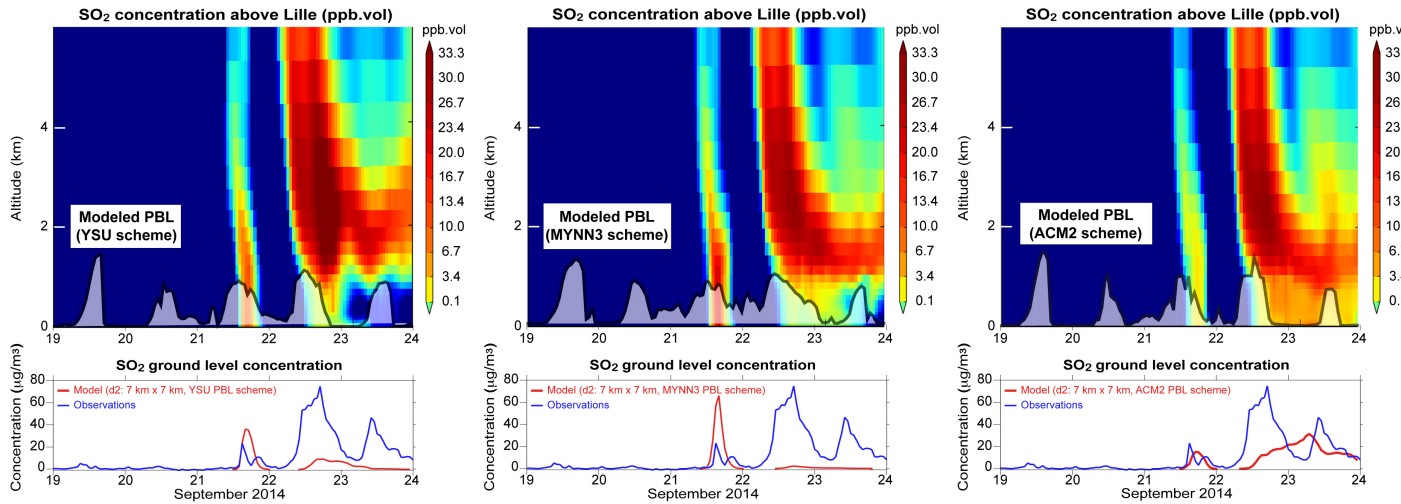

**Figure 14.** Comparison of far-range vertical distributions of $SO_2$ over Lille modeled with simulations at high spatial resolution configured with (left) YSU, (middle) MYNN3 and (right) ACM2 parameterization schemes of the planetary boundary layer. (Top) Modeled concentration of volcanic $SO_2$ as a function of time (X-axis) and altitude (Y-axis). The modeled PBL is overlaid in black. (Bottom) Time evolution of the observed (blue) and modeled (red) $SO_2$ concentrations at ground-level.

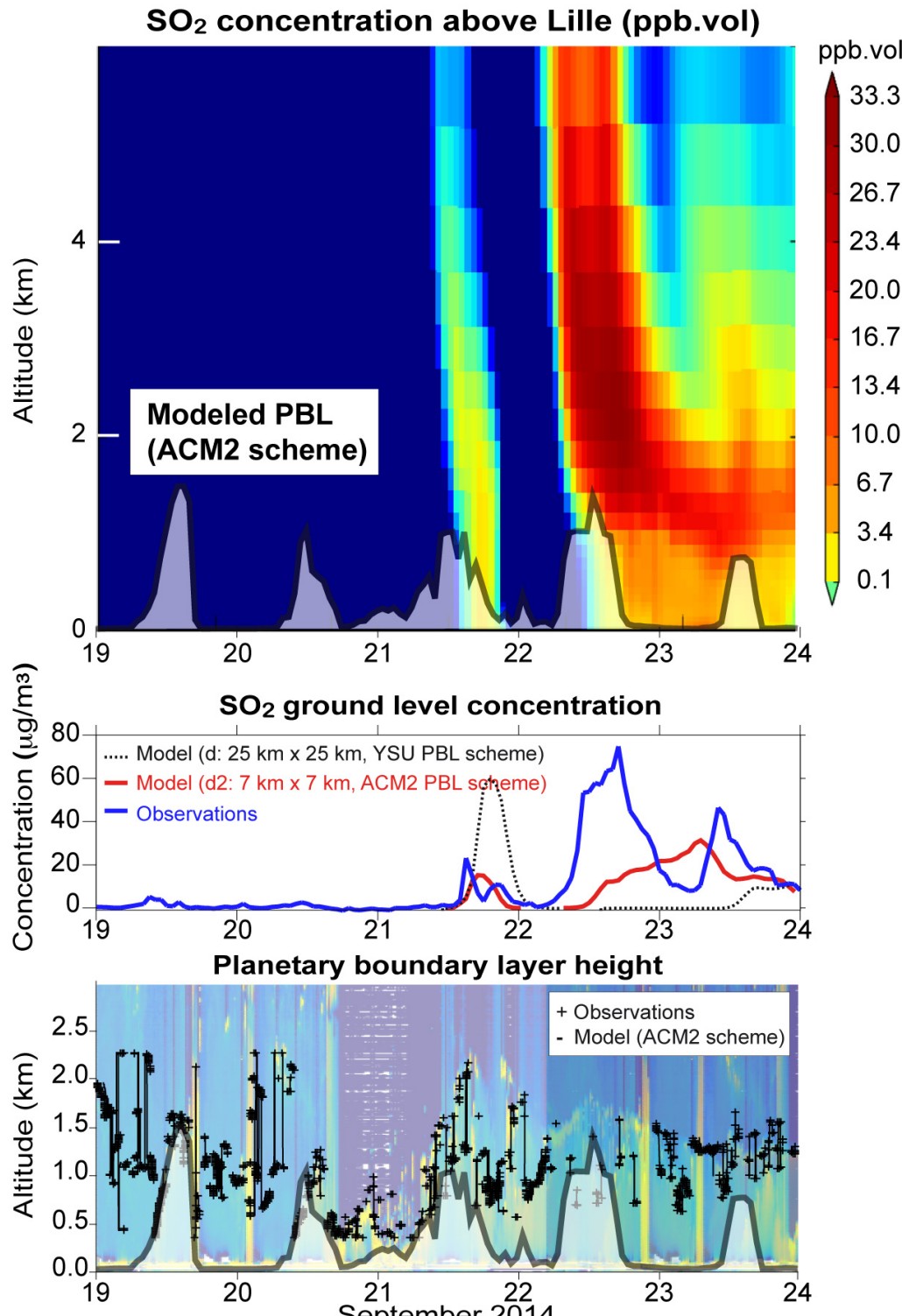

**Figure 15.** Key role of the PBL dynamics on far-range SO₂ concentrations at ground-level in Lille. (Top) Modeled concentration of volcanic SO₂ as a function of time (X-axis) and altitude (Y-axis) with simulations at high spatial resolution configured with ACM2 PBL parameterization scheme. The modeled PBL is overlaid in black. (Middle) Time evolution of the observed (blue) and modeled (red) SO₂ concentrations at ground-level. (Bottom) Comparison of modeled (black line) and observed (crosses) PBL height with time overlaid on lidar backscatter profile.