# Peer review of "Current challenges in modeling far-range air pollution induced by the 2014–15 Bárðarbunga fissure eruption (Iceland)"

_Atmospheric Chemistry and Physics, 2016_

## Referee Comment (RC1) · Anonymous Referee #1 · 13 Apr 2016

General comments

The paper by Boichu et al. presents a study on the far-range air pollution caused by the Badarbunga fissure eruption. The authors gather an extensive, and very useful, set of various type of measurements to complement and compare with modelling results. The paper is however rather descriptive and additional in depth evaluation of such an interesting dataset would be desirable. The authors encounter several problems in the modelling results that are not tackled. Although it is hard to tackle all of them, I would encourage the authors to suggest a roadmap on how to identify the main factors leading to a poor representation of the ground-level concentrations by the model. In addition, given that they have access to a full chemistry model, it would be worth to try to include $SO_2$ chemistry in the simulations.

[Figure]

Given the significance of the event, both in terms of air quality and also on volcanic emission forecasts and potential impacts, the paper shall be revised and considered for publication once the main aspects stated before and below are are addressed.

Specific comments

Introduction:

- Page 2 Line 10-11 " Even so, the Bardabunga eruption only weakly disturbed air traffic..." is unnecessary since aviation implications are not the topic of the paper and they are mostly significant for ash-rich eruptions. If the authors want to still keep the reference to aviation, they may state that "Whereas the Bardarbunga eruption $SO_2$ emissions were very large, but not constant, the ash emissions were limited and therefore no affectation in air traffic occurred, unlike on other occasions such as the Eyjafjallajokul eruption".

- Page 2 Line 12-13 "Nevertheless, Bardarbunga triggered a volcanogenic air pollution unprecedented in Europe... which necessitated locally exceptional civil protection measures". The sentence and posterior reference to Gisalson et al. 2015 (which only addresses the environmental stress in Iceland) is misleading since it reads as if exceptional civil protection measures were taken also in areas of Europe other than Iceland, please rephrase.

- Page 2 Line 17 "The Bardarbunga cloud travelled most often .... toward high latitudes". Please add reference, even if it is, for instance, Figure 4 of the manuscript.

- Page 2 Line 18 "peculiar meteorological conditions". They were not that peculiar given that, for instance, the Eyja event suffered similar transport conditions transporting the ash plume rapidly over mainland Europe. I would suggest "favourable conditions"

- Page 2 Lines 25 onwards until the end of the paragraph " Here, we use a wealth .... " is ambitious given that the paper is so far more descriptive and does not go in depth into the characterization of $SO_2$, the derived sulfates and the dynamics of the ABL

leading to such unusual concentrations at ground level. Please revisit this sentence after addressing the comments here presented and rephrase if needed. In addition, although the comparisons are indeed quantitatively, they require further analysis and better description in the text to be stated as it is now.

Methodology:

- Page 3 Line 8 "Given the low injection height" needs a reference.

-Page 3 Line 10 "The center of mass of the SO2 cloud is assumed to be within the PBL". What is this important statement based upon?

- Page 3 Line 20 The reference to the figure 1 should be complemented with additional explanation of the figure in the text. How this figure relates to the event the authors are examining? Are they suggesting that these low tropospheric aerosols are partly due to the event? What is the relation of the figure 1 with the topic of the paper?

- Page 3 Line 4 As in the previous comment, reference to figure 2. The figure is presented but all the information one can extract from the figure is not written in the text. Please do so and clearly stated how the figure relates with the influence of the volcanic eruption.

- Page 4 line 10 of "Chemistry-transport model". The authors state that the conversion from SO2 to SO4 is not implemented to avoid uncontrolled influence of uncertainties on the numerous factors governing this process in a volcanic cloud. It is unfortunate that the authors decided not to study the conversions since then the comparison with the aerosol measurements would have been more interesting. Given the characteristics of the eruption, with such a low height emission and transport, the conversions from SO2 to SO4 may be significant and one would hope that the CTM would at least reflect part of it. Have the authors at least tried to include the conversions? Given that the authors use a CTM, I would encourage them to add discussions on this and, if possible, an additional test with the conversion activated. Otherwise one may wonder why using

this model and not something closer to a Lagrangian particle dispersion model.

- Page 5 line 17. WRF can work using different PBL schemes. Is there any reason for using YSU in particular? Where there some sensitivity tests behind that suggested this one to be the one giving the best results? Given that the evolution of the PBL is crucial in this event to understand the ground level concentrations, more details on additional sensitivity tests, if done, would be useful and help understand the influence of this very important parameter in the final ground-level concentrations. Although not all the potential tests should be presented, for the sake of keeping the manuscript short, any insight in significant parameters is valuable.

- Page 5 Line 25 "inception time" what does this mean in this context?

- Page 5 Line 25 onwards: as for what I understand, the authors modified the injection height and times trying to match as much as possible the satellite data keeping a gaussian profile. Did they do this automatically or by simple visual inspection? It would be useful to know. It is also important to note that, the coarse assumptions in the source term make an accurate evaluation difficult. It would be good to highlight this in the conclusions section and state that the aim of the paper was not to make an estimate of the source term but to try to accommodate a simple source term that would represent the main features for this far-range study. A plot with the source term (injection height, times, vertical profiles) used in the modelling would be very useful to accompany figure 4 and would help the reader visualise the simulation.

- I would suggest also more description of Figure 4. For instance, we can clearly see from the derived IASI heights that for lower latitudes the heights are constrained to heights mostly below 8km. In addition, over many areas, example 20/09/2014 UK, the cloud is constrained below approximately 5 km a.s.l which will of course favour potential plume ground-touching.

Results:

As previously stated, if possible, it would be good to include a simulation that accounts for the SO2 conversions to sulfates.

- Section 3.1 title, Large scale SO2 dispersal from Iceland toward Europe does not read nice. I would suggest Large scale transport of SO2 towards Europe

- When looking at Figure 5, one has at least some doubts about the transport towards the Atlantic ocean of the Wave 1 since OMI show some traces that could actually be wave 1 transported further into mainland Europe. What is the opinion of the authors on this?

- Page 5 line 15-16, have the authors tried to gather data from the Scandinavia region to further assess the model behaviour in this region?

- Figure 6 c: why are the ground-level concentrations of SO2 and particles de-phased with particle concentrations peaking several hours after the passage of the SO2 plume? Whereas the text states there is coexistence of SO2 and sulfates, we see a delay in the peaking particle concentrations. We see this behaviour both for the first and second waves. Also, seeing the plots, it would be good to add a discussion of the PBL evolution and how this is may be influencing the concentrations at ground-level.

- Page 7 line 25 "Interestingly...". Why are the authors surprised about the two cities following a similar pattern? In sections before, the authors describe the transport patterns by explaining two waves coming towards Europe. This, therefore, makes it evident that the temporal patterns of the two locations may undergo a similar signal pattern. And actually the authors stated this right after the "Interestingly... " sentence. I would rephrase it and start with " As observed from space and reproduced by the CTM, two waves.... This is also seen in the measured ground-level concentrations at ..."

- The authors state that the model fails to represent the second wave. Looking at the magnitude of the model at the first wave I am wondering whether what actually happens is that the model is maybe too fast and representing the second wave too

early. Do the authors have any comments in this regard? Or, if not, do the authors have any suggestion on why the much significant peak is not at all captured by the model? Is it a transport problem? A mixing problem? A combination? Is it due to the assumptions in the source term? Given the discussion further on, it seems that the authors are, understandably, concerned about the representation of the PBL height. Have the authors made any tests in this regard? Also, as stated before, different PBL schemes in WRF can create different output. It would be good to have a clearer opinion of the authors on what factor they consider may be influencing most the poor model performance when representing the ground level concentrations and how would they approach a study to discern what is the main effect and how to compensate it (for example, as they have already suggested, increasing the resolution of the CTM and NWP calculations)

---

## Referee Comment (RC2) · Anonymous Referee #2 · 18 Apr 2016

General Comments

The paper 'Tracking far-range air pollution induced by the 2014–15 Bárdarbunga fissure eruption (Iceland)' describes a modelling exercise based on this particular eruption complemented by a large range of measurements. The paper is well written and well-structured and does a good job of highlighting notable and challenging aspects associated with this work, although there are a number of issues relating to the modelling aspect of the work that would need to be addressed before publication.

Specific comments

Discrepancies between models and observations are discussed and a number of reasons have been assigned to this. Possible explanations for these differences include:

[Figure]

Flux emission and altitude of injection 'This discrepancy can result from a limited knowledge of SO2 emission parameters (flux and altitude of injection) which initialize the chemistry-transport model.' It is also stated that 'Inception time and altitude of emissions are found by trial and error so as to reproduce first-order features of satellite and ground-level SO2 observations'. As the authors state model inversions can help with the refinement of this source term and help to further understand this discrepancy. This is clearly outside the scope of the work presented here although other possible reasons for the discrepancy may warrant further clarification. I think a more full discussion of SO2 oxidation and its possible contribution of the discrepancy should be included (after all it is a CTM). It is stated 'However, the conversion of SO2 to sulphate aerosols is not implemented in this study to avoid uncontrolled influence of uncertainties on the numerous factors governing this process in a volcanic cloud'. This is a reasonable approach although ground based measurements of sulphate aerosols suggest a fairly significant conversion which is not reflected in the source term. The inclusion of these interactions in future model iterations would clearly represent an improvement. Observations of the boundary layer heights compared to model simulations show a very large underestimation with the largest differences being observed at night time. The authors suggest that this is a ubiquitous feature of WRF. I would recommend confirming the influence of the boundary layer parameterisations by running WRF simulations using a number of parameterisations. This would confirm the influence of boundary layer height on the results presented here and may help to understand its contribution to model/observation mismatch. It is suggested that higher model resolution (temporal and spatial) may help elucidate further the source of observation/model differences and this has both further time and computational costs. This is a perfectly reasonable argument. However I do not think it would be not beyond the scope of this study to perform some test simulations at a higher resolution in order to shed light on this point.

In short I would suggest that perhaps a small effort in performing some simulations using a selection of boundary later parameterisations in WRF. Higher resolution simulations, if possible, would also help to strengthen (or at least clarify) some of the ideas

presented here. A more complete discussion of the SO2 oxidation should be also included. Exploring some other locations to confirm the model performance in other regions and add more credence to discussion and conclusions should be considered. Perhaps the authors might outline a possible framework for a set of simulations that might elucidate these uncertainties. The conclusion reiterates the issue surrounding the boundary layer in the model but this should be contextualised within the framework of the other possible reasons for model-observation mismatch.

Technical corrections

Page 1 Line 1 'has emitted'- is 'has' necessary?

Page 1 Line 3 'chemistry – transport' –model should be included after this for clarification

Page 2 Line 13' triggered a volcanogenic air pollution unprecedented'. Either 'a' should be removed or a descriptor after 'air pollution' should be included.

Page 4 Line 10 Do you need three references from the same author here?

Page 4 Line 12 This sentence regarding the omission of the SO2 chemistry could be improved. This will clearly lead to large uncertainties when comparing to SO2 mixing ratios. The measurements of the sulphate aerosols provide some information regarding the magnitude of the conversion process and should be included here

Page 5 Line 14 What was the spin time up on the WRF simulations?

Section 2.2 Line 24 What is the justification for choosing a Gaussian profile?

Section 3.1 Line 10 perhaps 'hitting' could be replaced with reaching

Figures

Figure 1 – It is hard to see how figure 1 is directly related to the text provided.

Figure 6c- Why might there a time shift between gas and aerosol?
* * *
Interactive
comment

Figure 9 – What would be an estimate of the uncertainty on the model boundary layer simulation?

Discussion - In the discussion the phrase 'finding optimum configuration' is used. This is something that could be undertaken or considered with the boundary layer parameterisation within WRF. This work would certainly strengthen some of the conclusions presented in this work.

---

## Author Comment (AC1) · 31 Jul 2016

The reviewers are thanked for their detailed and constructive reviews. As suggested by both reviewers, a substantial revision of the manuscript has been performed for exploring the impact on modeled far-range ground-level concentrations of 1) high spatial resolution model simulations (requiring an improvement of our computation capacities), 2) uncertainties on the source term and 3) the dynamics of the far-range planetary boundary layer.

These various tests allowed us to show the improvements in the modeled vertical distribution of the aged volcanic SO2 cloud reached with high-spatial resolution simulations. While variations in the altitude of SO2 injection at the source have a minor impact on far-range air quality modeling for this specific case-study, we show the key role played by the planetary boundary layer which is not accurately represented by state-of-the-art numerical weather prediction models.

Three new sections, including seven new figures, have consequently been added to the revised manuscript.

Answers to reviewers as well as changes made to the paper are detailed in the following in blue.

**Anonymous Referee #1**

General comments

The paper by Boichu et al. presents a study on the far-range air pollution caused by the Badarbunga fissure eruption. The authors gather an extensive, and very useful, set of various type of measurements to complement and compare with modelling results. The paper is however rather descriptive and additional in depth evaluation of such an interesting dataset would be desirable. The authors encounter several problems in the modelling results that are not tackled. Although it is hard to tackle all of them, I would encourage the authors to suggest a roadmap on how to identify the main factors leading to a poor representation of the ground-level concentrations by the model. In addition, given that they have access to a full chemistry model, it would be worth to try to include SO2 chemistry in the simulations.

Given the significance of the event, both in terms of air quality and also on volcanic emission forecasts and potential impacts, the paper shall be revised and considered for publication once the main aspects stated before and below are are addressed.

Specific comments

Introduction:

- Page 2 Line 10-11 " Even so, the Bardabunga eruption only weakly disturbed air traffic..." is unnecessary since aviation implications are not the topic of the paper and they are mostly significant for ash-rich eruptions. If the authors want to still keep the reference to aviation, they may state that "Whereas the Bardarbunga eruption SO2 emissions were very large, but not constant, the ash emissions were limited and therefore no affectation in air traffic occurred, unlike on other occasions such as the Eyjafjallajokul eruption". It has been rephrased accordingly (page 2, line 25-26).

- Page 2 Line 12-13 "Nevertheless, Bardarbunga triggered a volcanogenic air pollution unprecedented in Europe. . . which necessitated locally exceptional civil protection measures". The sentence and posterior reference to Gisalson et al. 2015 (which only addresses the environmental stress in Iceland) is

misleading since it reads as if exceptional civil protection measures were taken also in areas of Europe other than Iceland, please rephrase. It has been rephrased (page 2, line 28).

- Page 2 Line 17 "The Bardarbunga cloud travelled most often . . .. toward high latitudes". Please add reference, even if it is, for instance, Figure 4 of the manuscript. References to Fig. 4 (illustrating IASI SO2 altitude) and McCoy et al. (2015) GRL paper have been added (page 2, line 34).

- Page 2 Line 18 "peculiar meteorological conditions". They were not that peculiar given that, for instance, the Eyja event suffered similar transport conditions transporting the ash plume rapidly over mainland Europe. I would suggest "favourable conditions" this sentence was indeed confusing, it has been modified accordingly (page 2, line 34)

- Page 2 Lines 25 onwards until the end of the paragraph " Here, we use a wealth . . .. " is ambitious given that the paper is so far more descriptive and does not go in depth into the characterization of SO2, the derived sulfates and the dynamics of the ABL  leading to such unusual concentrations at ground level. Please revisit this sentence after addressing the comments here presented and rephrase if needed. In addition, although the comparisons are indeed quantitatively, they require further analysis and better description in the text to be stated as it is now.

We disagree with this comment, mentioning that this article is essentially descriptive, which seems to us very unfair.

Indeed, we first document this event by gathering and exploiting a large panel of volcanic SO2 observations from space (IASI, OMI), from ground-based remote sensing instruments (UV MAX-DOAS) and from in-situ ground-level air quality measurements at various locations. Moreover, we also gathered a vast panel of complementary observations of aerosols (sunphotometric, lidar and ground-level in-situ particulate measurement). The synergistic analysis of these various observations, using state-of-the-art retrieval algorithms, allowed us to provide a detailed characterisation of sulfate aerosols in a tropospheric volcanic cloud including timeseries of far-range vertical distribution, aerosol optical depth, volume size distribution with time and single scattering albedo. To our knowledge, such a detailed characterisation is rarely, if not never, available for a tropospheric plume compared to stratospheric plumes whose

lifetime is considerably longer rendering their observtion facilitated.

The combined analysis of both SO2 and sulfate aerosols allowed us to accurately describe the large-scale dispersal of the Bardarbunga cloud, its descent down to the lower troposphere and its arrival at the ground level.

Thanks to this detailed description of the volcanic cloud behaviour and additional observations of the far-range dynamics of the planteary boundary layer, a strong modeling effort was made to thoroughly test whether our modeling capacities are currently sufficient to simulate with accuracy this large-scale volcanogenic event of air pollution. After exploring various directions, our modeling study allowed therefore to show the various sources of difficulty encountered to reproduce correctly the magnitude of a volcanogenic pollution episode (spatial resolution of model simulations, uncertainty on source term, PBL dynamics). More precisely, this article (in its revised form) points out the key role played by the PBL dynamics and the limits of current state-of-the-art numerical weather prediction models for modeling it with sufficient accuracy for our case-study.

As such, we think that our paper brings indeed a wealth of complementary observations of SO2, sulfate aerosols and PBL dynamics. Such a panel of observations allowed us to perform a substantial modeling exercise and to highlight clearly the barriers which still need to be overcome in the coming years to accurately simulate and forecast such a volcanic vent in the coming future.

Methodology:- Page 3 Line 8 "Given the low injection height" needs a reference. Reference to Fig. 1 (previously Fig. 4 in the ACPD version), which shows an altitude of SO2 below 4-5 km near Iceland at the end of September 2014 and confirms the low injection height mentioned in the manuscript, has been added (page 3, line 20)

-Page 3 Line 10 "The center of mass of the SO2 cloud is assumed to be within the PBL". What is this important statement based upon?

The choice of the OMI SO2 product, between here PBL or TRL (lower troposphere) products which are associated respectively to a center of mass altitude of 0.9 and 2.5 km, has to be made according to independent information on the altitude of the volcanic cloud which are scarce for the

Bardarbunga eruption.

The chemistry-transport model allows for reproducing far-range ground-level concentrations assuming that a part of the emissions are injected at 1 km. Moreoever, IASI shows that the altitude of the $SO_2$ cloud near Iceland is below 4-5 km.

In this context, we tested both PBL and TRL products. The best agreement with model simulations (in terms of extent of the volcanic cloud) is reached with the OMI PBL product which is consequently chosen.

- Page 3 Line 20 The reference to the figure 1 should be complemented with additional explanation of the figure in the text. How this figure relates to the event the authors are examining? Are they suggesting that these low tropospheric aerosols are partly due to the event? What is the relation of the figure 1 with the topic of the paper?

The present reference to this figure in the text is indeed unclear as it seems to have been misunderstood. The text (page 4, lines 2-4) and caption of Figure 2 (previously Fig. 1) have been modified for clarification. This figure is useful to show that, while lidar observations are used to detect any kind of atmospheric particles, meteorological clouds can be clearly distinguished from aerosols (partly of volcanic origin here) in our case-study using adapted retrieval algorithms. Indeed, meteorological clouds evolve at a higher altitude that the aerosols that interest us lying at a low altitude (below 1.2 km).

- Page 3 Line 4 As in the previous comment, reference to figure 2. The figure is presented but all the information one can extract from the figure is not written in the text. Please do so and clearly state how the figure relates with the influence of the volcanic eruption.

For clarification, additional explanations on the exploitation of these data for characterizing volcanic aerosols have been added to the revised manuscript (page 4, lines 10-19).

- Page 4 line 10 of "Chemistry-transport model". The authors state that the conversion from $SO_2$ to $SO_4$ is not implemented to avoid uncontrolled influence of uncertainties on the numerous factors governing this process in a volcanic cloud. It is unfortunate that the authors decided not to study the

conversions since then the comparison with the aerosol measurements would have been more interesting. Given the characteristics of the eruption, with such a low height emission and transport, the conversions from SO2 to SO4 may be significant and one would hope that the CTM would at least reflect part of it. Have the authors at least tried to include the conversions? Given that the authors use a CTM, I would encourage them to add discussions on this and, if possible, an additional test with the conversion activated. Otherwise one may wonder why using this model and not something closer to a Lagrangian particle dispersion model.

The difficulty highlighted in our paper published in ACPD was to understand the reason why we missed entirely the second peak of ground-level SO2 concentration at all the studied monitoring stations. Including the conversion of SO2 gas to sulfate aerosols would not help to solve this problem. Indeed, the current absence of conversion leads only to an over-estimation of far-range SO2 concentrations and, subsequently, an overestimation of the SO2 emission flux (added page 5, lines 20-21). However, we have elucidated the reason for this problem in the revised manuscript by exploring the impact on ground-level concentrations of runninng model simulations at high spatial resolution, of the PBL dynamics, and of the source term (adding 3 new sections and 6 new figures).

As indicated by the reviewer, sulphates aerosols of volcanogenic origin may be important during such eruptions. Efforts will be made in the near future to implement this SO2 conversion into the chimere model. However, we think that a detailed validation study is also required prior to using the conversion scheme blindly. Taking advantage of the various observations that we gathered in this study and of the detailed characterization of sulfate aerosols that we performed, it represents the scope of another study.

- Page 5 line 17. WRF can work using different PBL schemes. Is there any reason for using YSU in particular? Where there some sensitivity tests behind that suggested this one to be the one giving the best results? Given that the evolution of the PBL is crucial in this event to understand the ground level concentrations, more details on additional sensitivity tests, if done, would be useful and help understand the influence of this very important parameter in the final ground-level concentrations. Although not all the potential tests should be presented, for the sake of keeping the manuscript short, any insight

in significant parameters is valuable.

YSU parameterization scheme has been used in initial simulations as it is the most widely used scheme in the WRF model. We tested in the revised manuscript two additional PBL parameterization schemes (ACM2 and MYNN3) and showed the key role played by the PBL dynamics to accurately simulate far-range ground-level SO2 concentrations in our case-study.

For this purpose, a new section (Section 4.4) was added including three new figures (Figures 13, 14 and 15). It shows how the PBL scheme controls the timing and amount of capture of the overlying volcanic SO2 cloud by the boundary layer, and subsequently the timing and magnitude of increase of the SO2 concentration at ground-level a few hours later. A large variability (up to ten-fold) of ground-level SO2 concentrations according to the chosen PBL scheme is highlighted.

Even if the ACM2 scheme provides the best fit to observations, none of the PBL schemes allows for rigorously modeling the second SO2 peak concentration with correct timing and intensity. We show that this difficulty likely results from the inaccuracy of the modeled PBL height time series. Indeed, this latter presents marked differences with observations retrieved from lidar observations, with a large underestimation of the modeled PBL height especially during mornings and evenings.

- Page 5 Line 25 "inception time" what does this mean in this context? This means 'time of release'. This has been modified in the text for clarification.

- Page 5 Line 25 onwards: as for what I understand, the authors modified the injection height and times trying to match as much as possible the satellite data keeping a gaussian profile. Did they do this automatically or by simple visual inspection? It would be useful to know. It is also important to note that, the coarse assumptions in the source term make an accurate evaluation difficult. It would be good to highlight this in the conclusions section and state that the aim of the paper was not to make an estimate of the source term but to try to accommodate a simple source term that would represent the main features for this far-range study. A plot with the source term (injection height, times, vertical profiles) used in the modelling would be very useful to accompany figure 4 and would help the reader visualise the simulation.

The objective of this study is not to provide an accurate description of source emissions but to show that a simple source term is indeed able to reproduce main features of a far-range volcanogenic air pollution event. For this reason, the best fit to satellite and ground-level SO2 concentrations was evaluated by visual inspection. This has been now better mentioned in the text (page 6 line 1).

The reconstruction of detailed emission timeseries for the whole duration of the eruption using inverse modeling tools, will be the subject of another paper in preparation.

An additional figure (Fig. 5), representing the simple source term used to initialize chemistry-transport simulations of the Bardarbunga SO2 cloud dispersal toward Europe, has been included in the revised manuscript.

- I would suggest also more description of Figure 4. For instance, we can clearly see from the derived IASI heights that for lower latitudes the heights are constrained to heights mostly below 8km. In addition, over many areas, example 20/09/2014 UK, the cloud is constrained below approximately 5 km a.s.l which will of course favour potential plume ground-touching.

A more thorough description of IASI images of the SO2 altitude was indeed lacking in the ACPD paper, as they allow for explaining a lot of features (such as the absence of the traces of SO2 over mainland Europe on 22 Sept) which are not reproduced by the model as they are associated to emissions released and then transported at high altitude toward regions (like northern Greenland, Fenno-Scandinavia) which are out of our domain of interest here (i.e. Western Europe). These descriptions have been added to Section 3.1 (page 6, lines 13-24).

Results:

As previously stated, if possible, it would be good to include a simulation that accounts for the SO2 conversions to sulfates. Answered earlier

- Section 3.1 title, Large scale SO2 dispersal from Iceland toward Europe does not read nice. I would suggest Large scale transport of SO2 towards Europe : title has been modified

- When looking at Figure 5, one has at least some doubts about the transport

towards the Atlantic ocean of the Wave 1 since OMI show some traces that could actually be wave 1 transported further into mainland Europe. What is the opinion of the authors on this?

An animation has been added to the revised manuscript (in the Supplementary Material), which shows more clearly than maps how Wave 1 is transported toward the Atlantic Ocean. According to IASI SO2 altitudes (Fig. 5), the SO2 traces detected by OMI on 22 Sept further above mainland Europe lie at a higher altitude (between 8 and 10 km asl) than Wave 1 (lying at an altitude below 4 km). According to IASI images of the SO2 altitude for the previous days, these traces are likely associated to a part of the the SO2 emissions released at a high altitude (above 8 km), transported first toward the Pole then dispersed over Fenno-Scandinavia. As stated in Section 2.2, these parcels of emissions, which do not travel toward Western Europe, are not taken into account in the modeled source-term.

Some explanations on these parcels of the volcanic SO2 cloud have been added to the revised manuscript (page 6, lines 13-24).

- Page 5 line 15-16, have the authors tried to gather data from the Scandinavia region to further assess the model behaviour in this region?

We have not tried to gather data in other regions as we could only gather for France (and nearby Belgium) a wealth of complementary observations on SO2, sulfate aerosols and PBL dynamics, the latter being particularly crucial to understand the key role played by the PBL and the source of discrepancies between model and observed far-range ground-level concentrations.

Moreover, according to us, the simple 2-wave behaviour of our french case-study, is an exceptional opportunity to assess our current modeling capacities.

Accordingly, adding additional observations is obviously of interest. Another study is underway to compare Eulerian vs. Lagrangian model approaches, as well as Era Interim/ECMWF vs. NCEP meteorological forcings. It will include many more stations in France, UK, Netherlands and Scandinavia.

- Figure 6 c: why are the ground-level concentrations of SO2 and particles de-phased with particle concentrations peaking several hours after the passage of the SO2 plume? Whereas the text states there is coexistence of SO2 and

sulfates, we see a delay in the peaking particle concentrations. We see this behaviour both for the first and second waves.

Indeed, SO2 and sulfate aerosols are more generally co-existent, e.g. a study of the Etna volcanic cloud by Boichu et al., ACP, 2015. Nevertheless, we need to mention that we do not have exhaustive observations to document thi co-existence as the signature of tropospheric sulfate aerosols of volcanic origin is difficult to isolate from the signature of co-existent of aerosols/particles of other type, especially in highly-polluted regions.

ACSM (aerosol chemical spectiation monitor) observations, which are not presented in this paper but included in another study in preparation, allow for characterizing the detailed chemistry of PM1 particles. They indicate that Bardarbunga SO2 and sulfate aerosols at ground-level are co-existent and follow a similar temporal pattern.

According to us, this shift between ground SO2 and particulate matter peaks consequently indicates the ground-level pollution by particles which are not only of volcanic origin. This result demonstrates the difficulty to rely only on ground-level PM observations, which do not provide information on the chemical signature of sampled aerosols, to identify and isolate the signature of aerosols of volcanic origin from co-existing aerosols of various possible origins especially at ground-level in a urban context which can be highly polluted.

Also, seeing the plots, it would be good to add a discussion of the PBL evolution and how this is may be influencing the concentrations at ground-level.

As mentioned above, an extensive exploration of the influence of various PBL parametererization schemes on far-range ground-level SO2 concentrations has been developed in the revised version of the paper (new Section 4.4) and allowed to show the crucial role played by the PBL dynamics to accurately model large-scale events of volcanogenic air pollution.

- Page 7 line 25 "Interestingly...". Why are the authors surprised about the two cities following a similar pattern? In sections before, the authors describe the transport patterns by explaining two waves coming towards Europe. This,

therefore, makes it evident that the temporal patterns of the two locations may undergo a similar signal pattern. And actually the authors stated this right after the "Interestingly. . . " sentence. I would rephrase it and start with " As observed from space and reproduced by the CTM, two waves. . .. This is also seen in the measured ground-level concentrations at ..."

These results seem not to be correclty presented in the ACPD version of the paper as the presence of a two-wave pattern is currently a result of our study combining model and observations, which allows for linking spatial to ground-based air quality observations thanks to model simulations.

We were indeed surprised to highlight such an interesting simple pattern, that we could follow in time progressing from the North to Central France at various ground monitoring stations. Compared to other studies performed in other regions (UK, Netherlands, Scandinavia) where air quality data seem to present a more 'chaotic' or 'disordered' behaviour (e.g. Ialongo et al., 2015; Schmidt et al., 2015), this french case-study seems consequently particularly interesting, as a kind of 'textbook case', for testing our current modeling capacities.

Text has been modified to clarify these points (page 8, lines 1-4).

- The authors state that the model fails to represent the second wave. Looking at the magnitude of the model at the first wave I am wondering whether what actually happens is that the model is maybe too fast and representing the second wave too early. Do the authors have any comments in this regard? Or, if not, do the authors have any suggestion on why the much significant peak is not at all captured by the model? Is it a transport problem? A mixing problem? A combination? Is it due to the assumptions in the source term? Given the discussion further on, it seems that the authors are, understandably, concerned about the representation of the PBL height. Have the authors made any tests in this regard? Also, as stated before, different PBL schemes in WRF can create different output. It would be good to have a clearer opinion of the authors on what factor they consider may be influencing most the poor model performance when representing the ground level concentrations and how would they approach a study to discern what is the main effect and how to compensate it (for example, as they have already suggested, increasing the resolution of the CTM and NWP calculations)

In the submitted ACPD paper, we suggested that the missing 2nd peak in ground-level concentrations resulted from an incorrect description of the vertical distribution of the volcanic cloud flying other the various monitoring ground stations. In this initial version of the article, we proposed as a future workplan to first develop simulations at a higher spatial resolution which could help to more accurately describe the long-range transport/dispersal of the volcanic cloud and its descent then its capture by the far-range boundary layer. We also suggested to explore the impact of the modeled PBL dynamics and the lack of a detailed knowledge of the source term.

In the revised manuscript, we explored these three hypotheses. For this purpose, we added three new sections (Sections 4.2, 4.3 and 4.4) which include six new figures (Figures 10 to 15).

First, we show that large improvements in air quality modeling are reached with simulations at higher horizontal resolution (Section 4.2). Initial simulations in the ACPD version were run using one single domain with a 25 km x 25 km horizontal grid resolution. After an upgrade of our computation capacities, we run, for the revised manuscript, simulations on two nested horizontal domains : the largest domain (from north Greenland down to Spain) with a coarse resolution of 22 km x 22 km and the narrower domain (large nevertheless from Norway down to Central France) with a finer resolution of of 7.3 km x 7.3 km.

We show how these high spatial resolution simulations allow for more accurately describe the vertical distribution of the far-range Bardarbunga cloud and its descent over France with time.

Despite clear improvements, discrepancies between modeled and observed SO2 concentrations at ground level remain.

Secondly, we introduce a variation in the flux and altitude of source emissions. For our specific case and period of study, we show that the source term plays a minor role and cannot solve the disagreement (Section 4.3).

Thirdly, as already mentioned above, we explore the impact of the PBL dynamics and show its key role to accurately model far-range voplcanogenic air pollution episode (Section 4.4). However, among the various PBL parameterization schemes that we tested, none of them allows for correctly

reproducing the right timing and intensity of capture of the overlying Bardarbunga cloud by the far-range boundary layer and subsequently the ground-level pollution episode. The substantial differences between modeled and observed (using lidar measurements) PBL height timeseries highlight the current limit of state-of-the-art mesoscale meterological models to solve this matter.

The paper 'Tracking far-range air pollution induced by the 2014–15 Bárdarbunga fis- sure eruption (Iceland)' describes a modelling exercise based on this particular erup- tion complemented by a large range of measurements. The paper is well written and well-structured and does a good job of highlighting notable and challenging aspects as- sociated with this work, although there are a number of issues relating to the modelling aspect of the work that would need to be addressed before publication.

Specific comments

Discrepancies between models and observations are discussed and a number of reasons have been assigned to this. Possible explanations for these differences include:

Flux emission and altitude of injection 'This discrepancy can result from a limited knowledge of SO2 emission parameters (flux and altitude of injection) which initialize the chemistry-transport model.' It is also stated that 'Inception time and altitude of emissions are found by trial and error so as to reproduce first-order features of satellite and ground-level SO2 observations'. As the authors state model inversions can help with the refinement of this source term and help to further understand this discrepancy. This is clearly outside the scope of the work presented here although other possible reasons for the discrepancy may warrant further clarification. I think a more full discussion of SO2 oxidation and its possible contribution of the discrepancy should be included (after all it is a CTM). It is stated 'However, the conversion of SO2 to sulphate aerosols is not implemented in this study to

avoid uncontrolled influence of uncertainties on the numerous factors governing this process in a volcanic cloud'. This is a reasonable approach although ground based measurements of sulphate aerosols suggest a fairly significant conversion which is not reflected in the source term. The inclusion of these interactions in future model iterations would clearly represent an improvement.

Exploration of the impact of uncertainties in the source term has been made in the revised version by testing various altitude of injection of SO2 emissions (from 3 to 7 km a.s.l.) and various emission flux values. We showed that in our specific case and time period of study, the source term plays a minor role on the far-range SO2 concentration at ground-level (new Section 4.3), compared to the spatial resolution of simulations (new Section 4.2) and the PBL dynamics (new Section 4.4).

Moreover, the objective of our paper is not to provide a detailed source-term for the Bardarbunga eruption. Reconstructing the Bardarbunga source term over the course of the whole eruption is nevertheless the goal of another paper in preparation developing inverse modeling procedures.

Our objective in the article here is to reproduce first order features of the far-range air pollution event triggered by this eruption using a simple source term. Hence, we showed that simulations at low spatial resolution with a simple source term do not allow for correctly representing this far-range pollution as the second large peak of SO2 concentration recorded at all monitoring stations is entirely missed.

The absence of conversion of SO2 to sulfate aerosols would not help to solve this problem. Indeed, this configuration only leads to an overestimation of the far-range SO2 abundance (as more SO2 should disappear at distance from the source by their conversion to sulfate which is currently not taken into account). Consequently, this process cannot help to reconciliate observations and model, where the modeled second peak of concentration is already substantially under-estimated (even completely missing).

Subsequently, this absence of conversion leads also to an over-estimation of the SO2 source term. However, reconstructing in detail the Bardarbunga source term in late September 2014 is not the goal of this paper. In addition, we think that validating the modeling of SO2 conversion to sulfate aerosols in

a tropospheric volcanic plume, with relevant observations of sulfate, is of crucial importance. Identifying and isolating the signature of sulfate aerosols of volcanic origin in a mix with aerosols of various types, as is commonly the case in polluted tropospheric regions, is also a challenge.

Therefore, taking advantage of the panel of volcanic sulfate observations gathered in this paper and their detailed characterization (time series of vertical distribution, aerosol optical depth, volume size distribution and single scattering albedo) retrieved here using state-of-the art algorithms, as well as additional observations not included in this article, we aim at developing a thorough validation of sulfur oxidation in a tropospheric volcanic plume using our chemistry-transport model in another article.

Observations of the boundary layer heights compared to model simulations show a very large underestimation with the largest differences being observed at night time. The authors suggest that this is a ubiquitous feature of WRF. I would recommend confirming the influence of the boundary layer parameterisations by running WRF simulations using a number of parameterisations. This would confirm the influence of boundary layer height on the results presented here and may help to understand its contribution to model/observation mismatch.

This point has also been raised by reviewer 1. For more details, please refer to the detailed answer made above. In a few words, we have indeed showed the key role played on the representation of far-range ground-level concentrations by the dynamics of the PBL (new Section 4.4). To do so, we have run multiple simulations for testing three PBL parameterization schemes (the most widely used YSU, but also ACM2 and MYNN3 schemes). These investigations showed that the ACM2 delivers the best fit to ground-level concentrations but some discrepancies remain on the timing (late of a few hours) and intensity (underevaluated by a factor of 3) of the second peak of concentration observed at far-range monitoring stations. The comparison of modeled PBL height timeseries with observations retrieved from lidar measurements allowed us to show that the inaccuracy in PBL model representation explains this shortcoming. Hence, this case-study illustrates how we reach here the limits of current state-of-the-art numerical weather prediction model, as the PBL dynamics represents one of the most challenging modeling task.

A specific section (Section 4.4) to develop this point, including 3 new figures, has been added to the revised version of the manuscript.

It is suggested that higher model resolution (temporal and spatial) may help elucidate further the source of observation/model differences and this has both further time and computational costs. This is a perfectly reasonable argument. However I do not think it would be not beyond the scope of this study to perform some test simulations at a higher resolution in order to shed light on this point.

This point has also been raised by reviewer 1. For more details, please refer to the detailed answer made above. In a few words, we have run simulations at higher horizontal and vertical resolutions which required first to improve our computation capacities (Section 4.2). Such new simulations are indeed highly ressource consuming as we run simulations with 2 nested horizontal grids. The narrower domain, extending nevertheless on a large region from Norway down to Central France, has a finer resolution of 7.3 km x 7.3 km. We showed how the higher horizontal resolution allowed for improving the long-range transport/dispersal of the volcanic cloud and the far-range vertical distribution of the volcanic SO2 cloud. Hence, the descent of the Bardarbunga SO2 cloud over France occurs earlier and at a higher speed than modeled with lower spatial resolution simulations. The volcanic cloud also touches now the ground on 22 Sept, which was not the case with initial simulations. Consecutive to these improvements, the model simulates a second peak in SO2 concentration, which was entirely missing with initial simulations at low resolution as the modeled Bardarbunga cloud was not descending down to the ground on 22 Sept.

A specific section (Section 4.2) to develop this point, including 3 new figures (Figures 10, 11 and 12), has been added to the revised version of the manuscript.

In short I would suggest that perhaps a small effort in performing some simulations using a selection of boundary later parameterisations in WRF. Higher resolution simulations, if possible, would also help to strengthen (or at least clarify) some of the ideas presented here. A more complete discussion of the SO2 oxidation should be also included.

The above-mentioned explorations of the impact of high-spatial resolution

simulations, PBL dynamics and source term have allowed to hierarchize the factors responsible for the discrepancies obtained between modeled and observed far-range ground-level concentrations. These investigations, developed in three new sections in the revised manuscript (Sections 4.2, 4.3 and 4.4), show that the source of model shortcomings mainly result from an inaccurate modeling of the PBL dynamics.

Exploring some other locations to confirm the model performance in other regions and add more credence to discussion and conclusions should be considered.

As mentioned earlier, the french case-study is especially relevant as we were able to highlight a regular pattern with two large peaks in SO2 concentration recorded at all ground-monitoring stations (and only separated by a time shift of a few hours). From a combined analysis of osbervations and model simulations, we showed that this specific behaviour results from the arrival of two volcanic SO2 waves over France. Compared to air quality observations published for other stations (in UK, Netherlands and Scandinavia) which present a more disordered behaviour, the regular dynamics of the french case-study represents a kind of textbook case, which seems to us particularly interesting for testing the accuracy of our model simulations.

Moreover, in addition to the exploitation of a large panel of SO2 and sulfate observations, we also have, for this french case-study, simultaneous information on the PBL dynamics retrieved from lidar observations. To our knowledge, such a rich panel of information is not available elsewhere and was crucial to understand and conclude on the crucial role played by the far-range PBL dynamics for robustly simulating volcanogenic air pollution events, and the current limitations of state-of-the-art NWP models for accurately modeling its dynamics.

Perhaps the authors might outline a possible framework for a set of simulations that might elucidate these uncertainties. The conclusion reiterates the issue surrounding the boundary layer in the model but this should be contextualised within the framework of the other possible reasons for model-observation mismatch.

Technical corrections

Page 1 Line 1 'has emitted'- is 'has' necessary? removed

Page 1 Line 3 'chemistry – transport' –model should be included after this for clarification added

Page 2 Line 13' triggered a volcanogenic air pollution unprecedented'. Either 'a' should be removed or a descriptor after 'air pollution' should be included. corrected

Page 4 Line 10 Do you need three references from the same author here? First and last references were only kept

Page 4 Line 12 This sentence regarding the omission of the SO2 chemistry could be improved. This will clearly lead to large uncertainties when comparing to SO2 mixing ratios. The measurements of the sulphate aerosols provide some information regarding the magnitude of the conversion process and should be included here

As explained above, not considering the conversion of SO2 to sulfate aerosols will lead to an over-estimation of SO2 source emissions to fit far-range SO2 abundance. This is not an issue as the objective of our study is not to provide an accurate but a first-order estimation of the Bardarbunga source in order to show that it can reproduce first-order features of the far-range SO2 column load and ground-level concentrations. These explanations have been added to the text of the revised manuscript (page 5, lines 19-21).

Page 5 Line 14 What was the spin time up on the WRF simulations?

The spin time up of WRF simulations is of five days. Added page 5, line 23.

Section 2.2 Line 24 What is the justification for choosing a Gaussian profile?

This choice has been made to point out the impact of a specific altitude of injection.

Section 3.1 Line 10 perhaps 'hitting' could be replaced with reaching replaced

Figures

Figure 1 – It is hard to see how figure 1 is directly related to the text provided.

As reviewer 1 made the same remark, the answer is already developed above. Text (page 4, lines 10-19) and caption of Fig. 2 (previously Fig. 1) have been consequently modified for clarifying this point.

Figure 6c- Why might there a time shift between gas and aerosol?

As developed for reviewer 1, ACSM observations (not included in this paper) providing a full chemistry description of PM1 component, shows that SO2 concentration is clearly correlated in time with sulfate aerosol concentration. From our point of view, this shift rather points out the mixing in the boundary layer of aerosols of various origins, not only volcanic. Contrary to SO2 which is a clear and unambiguous volcanic indicator, this shift shows the difficulty to extract the aerosol component of purely volcanic origin. This objective cannot be reached using only ground-level particulate matter observations. It requires more diverse measurements, such as photometric data explored in this paper which allow for fully characterizing sulfate aerosol optical depth, size distribution and single scattering albedo using state-of-the-art retrieval algorithms, in order to identify the signature of volcanic sulfate aerosols.

Figure 9 – What would be an estimate of the uncertainty on the model boundary layer simulation?

As already developed in answers to reviewer 1, many additional simulations for testing the impact of various WRF PBL parameterization schemes allowed us to estimate the associated uncertainty on modeled ground-level SO2 concentrations at a far distance from the volcano (see new Fig. 13 and 14). It also showed the limitations encountered with state-of-the-art NWP models as not any of these various PBL schemes is able to correctly reproduce the expected dynamics of the PBL as retrieved from lidar measurements (new Fig. 15).

Discussion - In the discussion the phrase 'finding optimum configuration' is used. This is something that could be undertaken or considered with the boundary layer parameterisation within WRF. This work would certainly strengthen some of the conclusions presented in this work.

As explained above, the impact on modeled far-range ground-level SO2 concentrations of both simulations performed at a higher spatial resolution (new section 4.2) and of various PBL WRF parametrisation schemes (new

section 4.4) has been developed in the revised manuscript.